# Towards Robust Influence Functions with Flat Validation Minima

Xichen Ye[1]  Yifan Wu[1]  Weizhong Zhang[1 2]  Cheng Jin[1 3]  Yifan Chen[4]

## Abstract

The Influence Function (IF) is a widely used technique for assessing the impact of individual training samples on model predictions. However, existing IF methods often fail to provide reliable influence estimates in deep neural networks, particularly when applied to noisy training data. This issue does not stem from inaccuracies in parameter change estimation, which has been the primary focus of prior research, but rather from deficiencies in loss change estimation, specifically due to the sharpness of validation risk. In this work, we establish a theoretical connection between influence estimation error, validation set risk, and its sharpness, underscoring the importance of flat validation minima for accurate influence estimation. Furthermore, we introduce a novel estimation form of Influence Function specifically designed for flat validation minima. Experimental results across various tasks validate the superiority of our approach.

## 1. Introduction

Training data is the fundamental component of machine learning, and its quality is critical for ensuring accurate and reliable predictions (Algan & Ulusoy, 2021; de Vargas et al., 2023). As large-scale foundation models, such as large language models (LLMs) (Brown et al., 2020) and text-to-image generative models (Rombach et al., 2022), become more prevalent, the demand for vast amounts of data to achieve high performance has grown exponentially. However, datasets—especially those sourced from the internet or generated by models—are often uncurated and may contain quite a few anomalous or mislabeled samples (Pleiss et al.,

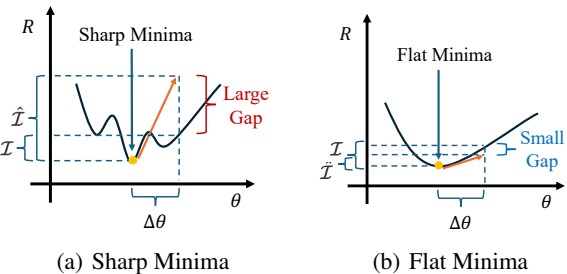

Figure 1. Illustration of flat minima. $R$-axis indicates the risk, while $\hat{\mathcal{I}}$ and $\mathcal{I}$ denote the estimated and actual influence (increase in empirical risk after deviating from minima), respectively. $\theta$ refers to the model parameters, and $\Delta\theta$ represents the parameter change. The orange arrows indicate the first-order or second-order approximation used to estimate the influence. As shown, the estimation gap is smaller in flat minima, leading to more reliable influence (marginal increase in risk) estimation.

2020). These issues degrade model performance, making it crucial to understand how training data and trained models interact (Hammoudeh & Lowd, 2024). One key aspect of this interaction is assessing the influence of individual training samples on model predictions (Koh & Liang, 2017).

A widely used approach for this purpose is the Influence Function (IF) (Koh & Liang, 2017), which provides insights into the predictions of Deep Neural Networks by estimating the impact of individual training samples. Specifically, IF estimates the change in the empirical risk on the validation set (or loss on a specific validation sample) resulting from small perturbations to the training set (e.g., the removal of a single training sample), using an estimated parameter change as a proxy. By serving as an efficient surrogate for the computationally expensive leave-one-out (LOO) retraining approach (Cook & Weisberg, 1982), IF enables the identification of the most influential samples in the training set of deep networks without retraining for each sample. Recent studies have successfully leveraged IF across various tasks, demonstrating its strong potential in practice (Pruthi et al., 2020; Han et al., 2020; Kong et al., 2022; Kim et al., 2023; Kwon et al., 2024).

Despite its broad applicability, existing Influence Functions often struggle when applied to noisy training data, posing a significant challenge since IF is supposed to distinguish high-quality data from noisy or anomalous sam-

[1]Fudan University [2]Shanghai Key Laboratory of Intelligent Information Processing [3]Innovation Center of Calligraphy and Painting Creation Technology, MCT, China [4]Hong Kong Baptist University. Correspondence to: Weizhong Zhang < weizhongzhang@fudan.edu.cn>, Yifan Chen <yifanc@hkbu.edu.hk>.

*Proceedings of the 42nd International Conference on Machine Learning*, Vancouver, Canada. PMLR 267, 2025. Copyright 2025 by the author(s).

ples. Specifically, we observe that regardless of whether a first-order (Pruthi et al., 2020) or second-order (Koh & Liang, 2017) approximation is used to estimate parameter change, the estimated influence tends to be ineffective once the model has converged on the noisy training set, as illustrated in Figure 2. This suggests that the limitation does not originate from inaccuracies in parameter change estimation, such as errors in inverse Hessian approximation, which have been the primary focus of previous studies (Koh & Liang, 2017; Kwon et al., 2024). Instead, we provide a novel understanding that the issue lies in the estimation of loss change, particularly the impact of local sharpness in the validation risk. Figure 1 provides an intuitive visualization of how local sharpness affects Influence Function estimation. As shown, even with a perfectly estimated parameter change $\Delta\theta$, a sharp local region can still introduce a substantial gap between the estimated and actual influence, ultimately undermining the reliability of influence estimation.

To this end, we introduce a second-order-based IF approximation on flat validation minima. This is motivated by both theoretical intuition (Figure 1) and empirical evidence (Figure 2), which highlight the critical role of flat validation minima in achieving precise influence estimation. We first establish a theoretical connection between the influence estimator and the validation set, recognizing the essential role of flat validation minima in achieving accurate influence estimation. Furthermore, we find that standard IF estimation becomes ineffective in optimization settings with flat minima (Figure 3). Our analysis attributes this limitation primarily to the zero-mean gradients in converged models and the misalignment in parameter change estimation (see Section 3.4 for more details). Considering this, we introduce a novel form of IF specifically designed for flat minima. This approach employs a second-order approximation to minimize the impact of vanishing gradients and introduces a refined parameter change estimation method tailored for flat test minima. We release the code at: `https://github.com/Virusdol/IF-FVM`.

We summarize our contributions as follows:

- We recognize the diminished effectiveness of influence estimators applied to training data containing noise and in optimization with flat minima.
- We establish general connections between the influence estimator and the validation set. We further devise a novel influence estimator specifically designed for network optimization with flat minima.
- Experimentally, we evaluate our proposed method across various tasks, demonstrating its superior performance.

## 2. Preliminaries

Consider a prediction problem from some input space $\mathcal{X}$ to an output space $\mathcal{Y}$. We are given an independently and identically (i.i.d.) sampled training dataset $S_{\text{tr}} = \{z_n = (x_n, y_n) \in \mathcal{X} \times \mathcal{Y}\}_{n=1}^{N}$. Empirical Risk Minimization (ERM) solves the following problem:

$$\theta^{\star} := \arg\min_{\theta} \frac{1}{N} \sum_{n=1}^{N} \ell(z_n, \theta), \tag{1}$$

where $\ell$ is the sample-wise loss function.

Given a validation set $S_{\text{val}} = \{z_m : (x_m, y_m)\}_{m=1}^{M}$, the Influence Function (IF) was introduced to answer *how a small perturbation $\epsilon$ to a training sample $z_{tr}$, specifically its removal, affects the prediction on a validation sample $z_{val}$.*

Intuitively, IF approximates the excess loss $\ell(z_{\text{val}}, \theta^{\star}_{z_{\text{tr}}}) - \ell(z_{\text{val}}, \theta^{\star})$ on a specific validation sample $z_{\text{val}}$, where $\theta^{\star}_{z_{\text{tr}}}$ denotes the minimizer of the risk after perturbing the empirical risk by assigning more weight to a certain training sample $z_{\text{tr}} \in S_{\text{tr}}$:

$$\theta^{\star}_{z_{\text{tr}}} := \arg\min_{\theta} \frac{1}{N} \sum_{n=1}^{N} \ell(z_n, \theta) + \epsilon\ell(z_{\text{tr}}, \theta), \tag{2}$$

where $z_n \in S_{\text{tr}}$ is the training sample. More specifically, the IF can be understood as a two-step approximation, as outlined below.

❶ The first step addresses the parameter change $\theta^{\star}_{z_{\text{tr}}} - \theta^{\star}$. Following Koh & Liang (2017), this parameter change can be approximated by a Newton step:

$$\theta^{\star}_{z_{\text{tr}}} - \theta^{\star} \approx -\epsilon H_{\text{tr}}^{-1} g_{z_{\text{tr}}}, \tag{3}$$

where $g_{z_{\text{tr}}} = \nabla_{\theta^{\star}}\ell(z_{\text{tr}}, \theta^{\star})$ represents the gradient of the loss with respect to $\theta^{\star}$ for the training sample $z_{\text{tr}}$, and $H_{\text{tr}} = \frac{1}{N}\sum_{n=1}^{N}\nabla_{\theta^{\star}}^{2}\ell(z_n, \theta^{\star})$ denotes the Hessian matrix computed over the training set $S_{\text{tr}}$, which is assumed to be positive definite (PD).

❷ The second step estimates the change in the loss $\ell(z_{\text{val}}, \theta^{\star}_{z_{\text{tr}}}) - \ell(z_{\text{val}}, \theta^{\star})$ of validation sample $z_{\text{val}}$ with respect to the parameter change $\theta^{\star}_{z_{\text{tr}}} - \theta^{\star}$. Using a first-order approximation, Koh & Liang (2017) approximate this change as:

$$\ell(z_{\text{val}}, \theta^{\star}_{z_{\text{tr}}}) - \ell(z_{\text{val}}, \theta^{\star}) \approx g_{z_{\text{val}}}^{\top}(\theta^{\star}_{z_{\text{tr}}} - \theta^{\star}), \tag{4}$$

where $g_{z_{\text{val}}} = \nabla_{\theta^{\star}}\ell(z_{\text{val}}, \theta^{\star})$.

Combining the two-step approximation, the Influence Function (IF) of the training sample $z_{\text{tr}}$ on the validation sample $z_{\text{val}}$ is defined as:

$$\mathcal{I}(z_{\text{tr}}; z_{\text{val}}) := g_{z_{\text{val}}}^{\top} H_{\text{tr}}^{-1} g_{z_{\text{tr}}}, \tag{5}$$

where the term $-\epsilon$ is omitted, as we are specifically interested in the removal effect of $z_{\text{tr}}$. This removal corresponds

to $\epsilon = -\frac{1}{N}$, making $-\epsilon$ a small positive constant that does not affect the relative influence.

Moreover, the Influence Function can also be extended to measure the overall effect on the full validation set $S_{\text{val}}$ as follows:

$$\mathcal{I}(z_{\text{tr}}; S_{\text{val}}) := \frac{1}{M} \sum_{m=1}^{M} g_{z_m}^{\top} H_{\text{tr}}^{-1} g_{z_{\text{tr}}}, \qquad (6)$$

where $z_m \in S_{\text{val}}$ represents a validation sample. This aggregated measure quantifies the impact of removing the training sample $z_{\text{tr}}$ on the overall validation set risk. For clarity, we refer to the IF defined in Equation (6) as the conventionally "**standard**" Influence Function from now on.

Based on the definition of Influence Function, a training sample $z_{\text{tr}}$ is said to have a **positive influence** on $z_{\text{val}}$ if $\text{sgn}(\mathcal{I}(z_{\text{tr}}, z_{\text{val}})) = 1$, meaning that the removal of $z_{\text{tr}}$ increases the loss value for the validation sample $z_{\text{val}}$. Conversely, $z_{\text{tr}}$ has a **negative influence** on $z_{\text{val}}$ if $\text{sgn}(\mathcal{I}(z_{\text{tr}}, z_{\text{val}})) = -1$, as its removal reduces the loss value for $z_{\text{val}}$. This concept of positive and negative influence can be generalized to evaluate the effect of $z_{\text{tr}}$ on the entire validation set $S_{\text{val}}$ by considering the aggregate influence defined by $\mathcal{I}(z_{\text{tr}}, S_{\text{val}})$. We note the sign choice is arbitrary in the literature (see a remark in Appendix G.1), up to the definition of loss change. Along this paper, we stick to the terminology above.

# 3. Flat Validation Minima for Influence Estimation

In this section, we first present the motivation behind a preliminary experiment reported in Figure 2 and Section 3.1. We then establish the theoretical link between influence estimation and validation set performance, demonstrating that the influence estimation error is upper-bounded by both the validation set risk and its sharpness (Section 3.2). Next, we show that the standard Influence Function, as defined in Equation (6), can be ineffective when applied to a flat validation set minima (Section 3.3). Finally, we introduce a novel Influence Function designed to overcome this limitation (Section 3.4).

## 3.1. Motivation

To clearly show the motivation for our investigation, we conduct a preliminary experiment on the mislabeled sample detection task. In this experiment, we track the performance of influence estimation alongside validation set accuracy throughout the training process. The results are presented in Figure 2.

As observed, the performance of influence estimation is highly correlated with validation set accuracy. This sug-

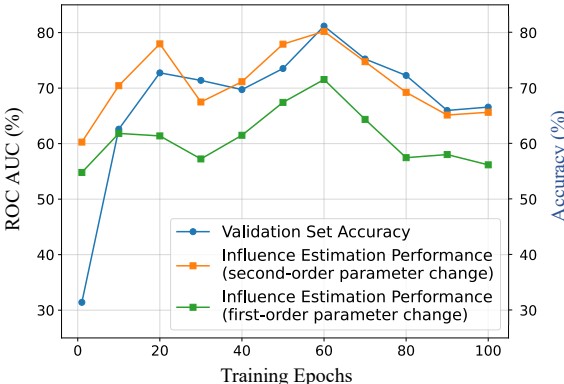

*Figure 2.* Validation set accuracy and influence estimation performance (measured by ROC AUC) for the mislabeled samples detection task across training epochs. The orange and green curves represent the performance of the Influence Function in identifying mislabeled samples, which are assumed to have a negative influence on the clean validation set. Influence is estimated using LiSSA (Koh & Liang, 2017) for second-order parameter change approximation and TracIn (Pruthi et al., 2020) for first-order parameter change approximation. As shown, influence estimation performance is highly correlated with validation set accuracy, regardless of whether the parameter change is estimated using a first-order or second-order approximation. The experiment is conducted on the CIFAR-10N (Wei et al., 2022) dataset under the "worst" setting. For further details, see Section 4.1.

gests that for an influence estimator to perform effectively, it should relate to a model that minimizes the risk on the validation set (as a model minimizing the population risk is inaccessible). Following this motivation, in the next subsection we explore the relationship between influence estimation and validation set performance.

## 3.2. Bound Influence Estimation Error from Above

To analyze the relationship between influence estimation and validation set performance, we begin by characterizing the performance of influence estimation from the perspective of risk minimization.

Specifically, we study whether the sign of the influence is correctly estimated. That is, given a training sample $z$ and a validation set $S_{\text{val}}$, we are interested in whether the sign of the estimated influence, $\text{sgn}(\mathcal{I}(z, S_{\text{val}}))$, matches the true influence sign $\text{sgn}(\mathcal{I}^{\star}(z, S_{\text{val}}))$; here, $\mathcal{I}^{\star}$ represents the target influence accurately capturing the true risk change associated with perturbations to the training sample. Based on this concept, we introduce the following definition for the error of influence estimation.

**Definition 3.1** (Influence Estimation Error)**.** Given an influence estimator $\mathcal{I}$, a target influence $\mathcal{I}^{\star}$, a validation set

$S_{\text{val}}$ and an underlying distribution $\mathcal{D}$, we define the random event:

$$A(z) := \{ \text{sgn}\,(\mathcal{I}(z, S_{\text{val}})) \neq \text{sgn}\,(\mathcal{I}^\star(z, S_{\text{val}})) \}. \quad (7)$$

The influence estimation error of $\mathcal{I}$ is defined as:

$$\mathcal{E}(\mathcal{I}) = \mathbb{E}_{z \sim \mathcal{D}} \left[ 1_{A(z)} \right] = \mathbb{P}_{z \sim \mathcal{D}} \left[ A(z) \right], \quad (8)$$

where $1_{\{\cdot\}}$ is the indicator function. Moreover, given a finite sample set $S = \{z_n\}_{n=1}^N$, the empirical influence estimation error of $\mathcal{I}$ is defined as:

$$\hat{\mathcal{E}}_S(\mathcal{I}) = \frac{1}{N} \sum_{n=1}^N \left[ 1_{A(z_n)} \right]. \quad (9)$$

Definition 3.1 provides a framework for quantifying the quality of an influence estimator. Based on this, we establish the following influence estimation error bound.

**Theorem 3.2** (Upper Bound on Generalization Influence Estimation Error). *Given an influence estimator $\mathcal{I}$ in a certain function space $\mathcal{H}$, a target influence estimator $\mathcal{I}^\star$, a fixed validation set $S_{\text{val}}$, and an underlying distribution $\mathcal{D}$. We then condition the random variable $z \sim \mathcal{D}$ on the influence signs to induce two mixture components of the distribution $\mathcal{D}$:*

$$\mathcal{D}_+ := \mathbb{P}_{z|\{sgn(\mathcal{I}^\star(z,S_{val}))=1\}}, \quad \mathcal{D}_- := \mathbb{P}_{z|\{sgn(\mathcal{I}^\star(z,S_{val}))=-1\}}.$$

*If the following conditions hold:*

$$\mathbb{E}_{z \sim \mathcal{D}_+}[\mathcal{I}(z, S_{val})] > 0 \ \text{ and } \ \mathbb{E}_{z \sim \mathcal{D}_-}[\mathcal{I}(z, S_{val})] < 0, \quad (10)$$

*then the generalization influence estimation error $\mathcal{E}(\mathcal{I})$ is upper bounded by:*

$$\mathcal{E}(\mathcal{I}) \leq \exp\left( -\frac{2\mu^2}{\hat{R}_{val}^\gamma(\theta)^2} \right), \quad (11)$$

*where $\mu = \inf_{(\mathcal{I}, \mathcal{D}') \in \mathcal{H} \times \{\mathcal{D}_+, \mathcal{D}_-\}} |\mathbb{E}_{z \sim \mathcal{D}'}[\mathcal{I}(z, S_{val})]|$, and*

$$\hat{R}_{val}^\gamma(\theta) := \max_{\|\Delta\| \leq \gamma} \hat{R}_{val}(\theta + \Delta), \quad (12)$$

*with $\gamma = \sup_{z \sim D} \|\theta_z - \theta\|$, where $\theta$ is the model parameter.*

**Remark**. The assumptions $\mathbb{E}_{z \sim \mathcal{D}_+}[\mathcal{I}(z, S_{\text{val}})] > 0$ and $\mathbb{E}_{z \sim \mathcal{D}_-}[\mathcal{I}(z, S_{\text{val}})] < 0$ are fairly mild, as they simply require the overall influence estimation performance to be better than random guessing.

Building on Theorem 3.2, the result can be readily extended to the case where we only have finite samples from $\mathcal{D}$. This extension is formalized in the following corollary.

**Corollary 3.3** (Upper Bound on Empirical Influence Estimation Error). *Given an influence estimator $\mathcal{I}$, a target influence estimator $\mathcal{I}^\star$, a validation set $S_{val}$, and a set $S = \{z_n\}_{n=1}^N$ where each sample $z_n$ is i.i.d. drawn from an underlying distribution $\mathcal{D}$. If the assumptions in Theorem 3.2 hold, then with probability at least $1 - \delta$, the empirical influence estimation error $\hat{\mathcal{E}}_S(\mathcal{I})$ is upper bounded by:*

$$\hat{\mathcal{E}}_S(\mathcal{I}) \leq \exp\left( -\frac{2\mu^2}{\hat{R}_{val}^\gamma(\theta)^2} \right) + \sqrt{-\frac{\log \delta}{2N}}. \quad (13)$$

The proof of Theorem 3.2 and Corollary 3.3 are provided in Appendices D.1 and D.2, respectively.

### 3.3. Flat Validation Minima via Sharpness-Aware Minimization

Theorem 3.2 and Corollary 3.3 suggest that, under mild assumptions, the risk of influence estimation error can be effectively controlled by managing $R_{\text{val}}^\gamma(\theta)$. This implies that influence should be computed on $\tilde{\theta}$, which is obtained by solving the following optimization problem:

$$\tilde{\theta} := \arg\min_\theta \hat{R}_{\text{val}}^\gamma(\theta), \quad (14)$$

where $\gamma$ is a fixed constant, and $R_{\text{val}}^\gamma(\theta)$ is defined as:

$$\hat{R}_{\text{val}}^\gamma(\theta) := \max_{\|\Delta\| \leq \gamma} \hat{R}_{\text{val}}(\theta + \Delta) \quad (15)$$

$$= \underbrace{\left[ \max_{\|\Delta\| \leq \gamma} \hat{R}_{\text{val}}(\theta + \Delta) - \hat{R}_{\text{val}}(\theta) \right]}_{\text{sharpness}} + \hat{R}_{\text{val}}(\theta). \quad (16)$$

While Equation (15) differs from the standard objective of Empirical Risk Minimization (ERM), this optimization problem can be effectively addressed using existing techniques.

Specifically, this formulation naturally aligns with Sharpness-Aware Minimization (SAM) (Foret et al., 2021), a learning paradigm distinct from ERM. Instead of directly minimizing the empirical risk, SAM simultaneously minimizes both the loss value and the loss sharpness, aiming to find a flat minima. Intuitively, a flat minima leads to a model with better robustness and generalization ability.

However, empirically, we find that the standard Influence Function, as defined in Equation (6), can be ineffective when optimizing Equation (15). To highlight this, we track the estimated influence during the optimization of $\hat{R}_{\text{val}}^\gamma(\theta)$, with the results shown in Figure 3(a). As illustrated, the performance of the standard Influence Function deteriorates as $\hat{R}_{\text{val}}^\gamma(\theta)$ decreases.

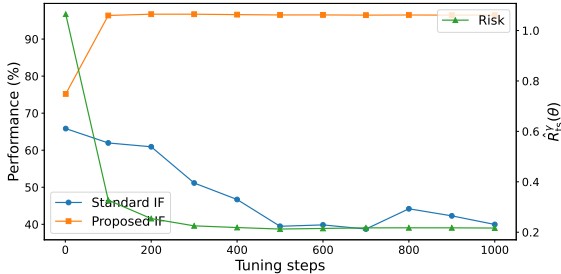

(a) Influence Estimation Performance Across Tuning Steps

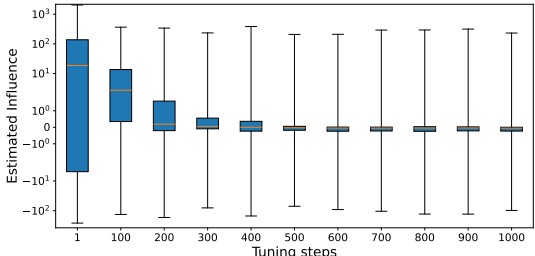

(b) Box Plot of Estimated Influence for Clean Samples Using the Standard IF Across Tuning Steps

*Figure 3.* (a) The influence estimation performance, measured by ROC AUC, for the mislabeled sample detection task across tuning steps. The standard influence is estimated using LiSSA (Koh & Liang, 2017), with SAM (Foret et al., 2021) employed as the flat minima solver. As shown, the performance of the standard Influence Function decreases as $\hat{R}_{val}^{\gamma}(\theta)$ decreases, while our proposed Influence Function shows an improvement. The experiment is conducted on the CIFAR-10N (Wei et al., 2022) dataset under the "worst" setting. For more details, see Section 4.1. (b) Box plot of influence for clean samples estimated using the standard Influence Function across tuning steps. As illustrated, the absolute value of the estimated influence continuously decreases.

Figure 3(b) further uncovers the underlying limitations of the standard Influence Function. Specifically, during the tuning process, the estimated influence of clean samples consistently approaches zero. This behavior directly relates to our analysis of the error bound in influence estimation. Recall the definition of $\mu$, where $\mu = \inf_{(\mathcal{I}, \mathcal{D}') \in \mathcal{H} \times \{\mathcal{D}_+, \mathcal{D}_-\}} |\mathbb{E}_{z \sim \mathcal{D}'} [\mathcal{I}(z, S_{val})]|$. As the estimated influence of clean samples $\mathbb{E}_{z \sim \mathcal{D}_+} [\mathcal{I}(z, S_{val})]$ approaches zero, we observe that $\mu \to 0$, which lifts the upper bound and allows influence estimation error increase.

To address this, we introduce a novel Influence Function in the following section, enabling the computation of sample influence on flat validation set minima.

### 3.4. Influence Function in Flat Validation Minima

**Limitations of the Standard Influence Function**. We identify two key limitations of the standard Influence Function when applied to flat validation minima: inaccuracies

in estimating both ❶ the parameter change and ❷ the loss change.

❶ For parameter change, the standard IF assumes that the model parameter minimizes the risk on the training set. However, this assumption does not align with the flat validation minima framework, as the current minima are derived from the validation set rather than the training set. This misalignment introduces a discrepancy that inherently compromises the accuracy of parameter change estimation.

❷ For loss change, the standard IF utilizes a first-order approximation term for estimation. However, previous studies (Kim et al., 2023) have shown that the gradients of pre-trained neural networks can be approximated as a Gaussian distribution centered at zero. This property naturally causes the estimated influence, $\mathbb{E}_{z_{tr} \in S_{tr}}[\mathcal{I}(z_{tr}, S_{val})] = \mathbb{E}_{z_{tr} \in S_{tr}}[g_{val}^\top H_{tr}^{-1} g_{z_{tr}}]$, to approach zero. Consequently, this leads to $\mu \to 0$, which severely limits the effectiveness of the standard estimation.

**Proposed Influence Funtion**. To address the two aforementioned limitations, we introduce a novel Influence Function incorporating newly designed terms for ❶ parameter change and ❷ loss change.

❶ We begin by examining the parameter change. Rather than focusing on $\theta_{z_{tr}}^\star$ as described in Section 2, we consider $\tilde{\theta}_{z_{tr}}$, defined as:

$$\tilde{\theta}_{z_{tr}} := \arg\min_\theta \max_{\|\Delta\| \leq \gamma} \hat{R}_{val}(\theta + \Delta) + \epsilon\ell(z_{tr}, \theta + \Delta). \quad (17)$$

Here, $\tilde{\theta}_{z_{tr}}$ denotes the perturbed parameter resulting from incorporating the training sample $z_{tr}$ into the optimization objective, i.e., Equation (15), with a small perturbation $\epsilon$.

Similar to standard IF, the parameter change $\tilde{\theta}_{z_{tr}} - \tilde{\theta}$ can be approximated using a Newton step:

$$\tilde{\theta}_{z_{tr}} - \tilde{\theta} \approx -\epsilon\tilde{H}_{val}^{-1}\tilde{g}_{z_{tr}} \quad (18)$$

where $\tilde{g}_{z_{tr}} = \nabla_{\tilde{\theta}}\ell(z_{tr}, \tilde{\theta}_{z_{tr}})$ represents the gradient of the loss with respect to $\tilde{\theta}$ for the training sample $z_{tr}$, and $\tilde{H}_{val} = \frac{1}{M}\sum_{m=1}^M \nabla_{\tilde{\theta}}^2\ell(z_m, \tilde{\theta})$ denotes the Hessian matrix computed over the validation set $S_{val}$, which is assumed to be positive definite (PD). A detailed derivation is provided in Appendix E.1.

❷ Next, we address the issue of previous first-order estimation methods via introducing second-order approximation. Consider the loss change on a validation sample $z_{val}$, and we denote the parameter change $\Delta\theta = \tilde{\theta}_{z_{tr}} - \tilde{\theta}$. We propose to estimate the loss change with respect to the parameter changes $\Delta\theta$ using a second-order approximation term:

$$\ell(z_{val}, \tilde{\theta}_{z_{tr}}) - \ell(z_{val}, \tilde{\theta}) \approx \frac{1}{2}\Delta\theta^\top \nabla_{\tilde{\theta}}^2 z\ell(z_{val}, \tilde{\theta})\Delta\theta, \quad (19)$$

where $\tilde{g}_{z_{val}} = \nabla_\theta\ell(z_{val}, \tilde{\theta})$.

*Table 1.* Area Under the Receiver Operating Characteristic Curve (ROC AUC) (%) and Average Precision (AP) (%) for identifying mislabeled samples on the CIFAR-10N/-100N datasets across different influence estimation methods. Results (mean±std) are reported over 3 random runs. **Bold** indicates the best results and underline denotes the second-best results.

| Method | CIFAR-10N | | | | | | CIFAR-100N | |
| | Aggre | | Random | | Worst | | Noisy | |
| | ROC AUC | AP | ROC AUC | AP | ROC AUC | AP | ROC AUC | AP |
|---|---|---|---|---|---|---|---|---|
| LiSSA | 59.74±2.91 | 33.45±1.75 | 59.78±2.77 | 45.01±2.70 | 65.75±0.39 | 68.35±0.74 | 57.48±1.70 | 51.28±0.87 |
| LiSSA* | 63.70±1.09 | 49.58±0.99 | 73.48±1.90 | 67.81±2.23 | 78.96±1.46 | 81.31±1.32 | 53.95±1.59 | 56.54±1.08 |
| TracIn | 53.91±5.85 | 28.94±5.10 | 61.61±0.74 | 43.83±1.79 | 65.74±2.32 | 62.91±1.75 | 56.13±2.51 | 52.34±2.62 |
| GEX | 87.38±1.21 | 78.01±1.05 | 91.11±0.53 | 87.57±0.42 | 93.28±0.10 | 94.16±0.10 | 90.17±0.70 | **89.05±0.53** |
| DataInf | 58.50±3.98 | 30.53±1.62 | 54.50±2.32 | 38.15±2.73 | 55.49±1.45 | 57.14±1.53 | 53.69±1.35 | 48.73±1.10 |
| DataInf* | 63.31±1.44 | 49.98±1.92 | 73.85±3.33 | 68.57±4.65 | 81.36±1.98 | 83.99±1.69 | 50.95±1.14 | 53.74±0.48 |
| VM | 95.18±0.15 | 76.31±0.21 | 95.92±0.10 | 87.35±0.45 | 95.88±0.13 | 94.27±0.21 | 89.77±0.08 | 83.81±0.18 |
| FVM | **96.14±0.06** | **79.53±0.27** | **96.63±0.03** | **88.82±0.26** | **96.46±0.08** | **94.97±0.12** | **90.80±0.04** | 85.41±0.05 |

**Combining Equations ([18](#)) and ([19](#))**, we define the influence on the flat validation minima $\tilde{\theta}$ as follows:

$$\mathcal{I}(z_{\text{tr}}, z_{\text{val}}) := \tilde{g}_{z_{\text{val}}} \tilde{H}_{\text{val}}^{-1} \tilde{g}_{z_{\text{tr}}} \\ + \frac{1}{2} \epsilon \tilde{g}_{z_{\text{tr}}}^{\top} \tilde{H}_{\text{val}}^{-1} \nabla_{\tilde{\theta}}^2 \ell(z_{\text{val}}, \tilde{\theta}) \tilde{H}_{\text{val}}^{-1} \tilde{g}_{z_{\text{tr}}}. \tag{20}$$

Furthermore, this Influence Function can be extended to measure the overall effect on the entire validation set $S_{\text{val}}$:

$$\mathcal{I}(z_{\text{tr}}, S_{\text{val}}) := \tilde{g}_{z_{\text{tr}}}^{\top} \tilde{H}_{\text{val}}^{-1} \tilde{g}_{z_{\text{tr}}}. \tag{21}$$

A detailed derivation is provided in Appendix [E.2](#) and detailed implementations are outlined in Appendix [A](#).

**Validation Minima v.s. Flat Validation Minima**. Our theoretical analysis suggests that flat validation minima are essential for better influence estimation performance. However, in practice, directly optimizing $\hat{R}_{\text{val}}^{\gamma}(\theta)$ can be computationally expensive. In such cases, the validation minima $\tilde{\theta} := \arg\min_{\theta} \hat{R}_{\text{val}}(\theta)$ can serve as a surrogate for the flat validation minima. We denote our proposed Influence Function for Validation Minima as VM and Flat Validation Minima as FVM. In the following experiments, we evaluate both methods.

**Remark**: Unlike the standard influence function, which assigns positive scores to useful samples and negative scores to harmful ones, our influence function exclusively produces positive scores. Nevertheless, the distribution of influence scores generated by our method is sufficiently distinct to identify influential samples.

## 4. Experiments

In this section, we conduct experiments across various tasks to evaluate the effectiveness of our proposed approach. In Section [4.1](#), we apply our method to the mislabeled sample detection task, demonstrating its ability to accurately

*Table 2.* Top-1 relabeling accuracy (%) on CIFAR-10N dataset for different influence estimation methods. Results (mean±std) are reported over 3 random runs. **Bold** indicates the best results and underline denotes the second-best results.

| Method | Aggre | Random | Worst |
|---|---|---|---|
| LiSSA | 5.28±3.15 | 9.04±2.70 | 19.32±2.59 |
| LiSSA* | 31.33±8.97 | 75.87±4.54 | 71.47±1.71 |
| TracIn | 37.08±4.99 | 53.28±8.86 | 50.66±0.92 |
| GEX | 30.19±3.95 | 54.03±4.31 | 80.35±1.92 |
| DataInf | 5.59±2.20 | 22.24±5.41 | 21.25±7.36 |
| DataInf* | 40.58±14.48 | 81.57±4.07 | 76.19±0.87 |
| VM | 94.17±0.02 | 91.94±0.08 | 85.01±0.35 |
| FVM | **94.63±0.04** | **92.46±0.07** | **86.09±0.38** |

identify mislabeled samples. In Section [4.2](#), we validate its effectiveness in relabeling samples within a noisy training set. In Section [4.3](#) and Section [4.4](#), we assess our method's capability in identifying influential samples in text and image generation tasks. In Section [4.5](#), we demonstrate the significance of our designed components.

### 4.1. Mislabeled Sample Detection

In this section, we evaluate the performance of influence approximation methods in identifying mislabeled (noisy label) samples. This task is based on the assumption that correctly labeled samples contribute positively to reducing the risk on the clean validation set, while mislabeled samples exhibit adverse effects. Under this assumption, the estimated influence $\mathcal{I}(z, S_{\text{val}})$ for mislabeled samples is expected to be lower than that of correctly labeled samples. Specifically, we treat mislabeled samples as the positive class to be detected and assess detection performance using Area Under the Receiver Operating Characteristic Curve (ROC AUC), and Average Precision (AP) metrics.

*Table 3.* Top-1 relabeling accuracy (%) on CIFAR-100N dataset for different influence estimation methods. Results (mean±std) are reported over 3 random runs. **Bold** indicates the best results and underline denotes the second-best results.

| Method | LiSSA | LiSSA* | TracIn | GEX | DataInf | DataInf* | TM | FVM |
|---|---|---|---|---|---|---|---|---|
| Acc | 0.28±0.07 | 2.55±0.92 | 20.11±2.50 | 22.41±0.46 | 1.24±0.56 | 5.81±2.70 | 58.13±0.15 | **70.61±0.24** |

We compare our proposed methods against several influence estimation approaches, including LiSSA (Koh & Liang, 2017), TracIn (Pruthi et al., 2020), GEX (Kim et al., 2023), and DataInf (Kwon et al., 2024). Additionally, we compare our methods with modified versions of LiSSA and DataInf, computed using the checkpoint from the training process that achieves the best results, referred to as LiSSA* and DataInf*. The experiments are conducted on CIFAR-10/-100N datasets (Wei et al., 2022). Detailed descriptions of the experimental setup can be found in Appendix B.1.

The results, presented in Table 1, demonstrate that our proposed methods, particularly FVM, outperform existing influence estimation techniques across all noise levels and settings. The only exception is the AP of GEX on CIFAR-100N under the "Noisy" setting. However, GEX requires extensive tuning, specifically 32 times, to achieve optimal performance, which is computationally expensive. When we limit the tuning of GEX to a single time, comparable to our method, its AP drops to 84.26, which is lower than that of our approach. These experimental results validate the superiority of our proposed methods in detecting mislabeled samples, highlighting that optimizing for flat validation minima is crucial for achieving more accurate and reliable influence estimation.

### 4.2. Training Sample Relabeling

Following up the detection task in Section 4.1, we extend our analysis to the sample relabeling task. This task is based on the further assumption that the correct class for a training sample has the most significant influence on reducing the risk of the clean validation set. Under this assumption, given a training sample $z_n = (x_n, y_n)$, we define the relabeling function as follows:

$$\hat{y}_n = \arg\max_k \mathcal{I}\left((x_n, k), S_{\text{val}}\right), \quad (22)$$

where $k \in \{1, \cdots, K\}$ represents the set of possible classes. To evaluate the relabeling performance, we compute the top-1 accuracy by comparing the true label $y_n$ with the predicted label $\hat{y}_n$.

The results are presented in Table 2 and Table 3. As shown, our proposed methods, VM and FVM, significantly outperform existing approaches by a large margin. Additionally, the results on CIFAR-10N highlight the importance of flat validation minima, with FVM achieving a 12% improve-

*Table 4.* ROC AUC (%) and Recall (%) for identifying influential samples in text generation tasks across different influence estimation methods. **Bold** indicates the best results and underline denotes the second-best results. The results for TracIn and DataInf were generated by executing the official implementation provided by DataInf. According to DataInf (Kwon et al., 2024), obtaining results around 99.99% is considered reasonable for this experiment.

| Task | Method | ROC AUC | Recall |
|---|---|---|---|
| Sentence Transformations | TracIn | 94.95±6.14 | 70.97±25.18 |
| | DataInf | 99.58±1.96 | 96.18±9.33 |
| | VM | **99.97±0.16** | **99.10±2.79** |
| | FVM | 99.92±0.18 | 98.47±2.90 |
| Math Problems (w/ Reasoning) | TracIn | 78.50±17.77 | 26.61±39.95 |
| | DataInf | 99.86±0.68 | 97.37±6.97 |
| | VM | 99.96±0.21 | 98.86±3.28 |
| | FVM | **99.99±0.07** | **99.38±1.65** |

ment over VM. These findings underscore the critical role of seeking flat minima for enhancing influence estimation performance.

### 4.3. Influential Sample Identification in Text Generation

To further demonstrate the effectiveness of our proposed method, we evaluate its ability to identify influential samples in text generation, following Kwon et al. (2024). The assumption for this task is analogous to that in the mislabeled sample detection task (Section 4.1): for a generated sample, we assume that training samples from the same class contribute the most to the generation, while samples from other classes contribute less. Consequently, training samples that share the same class as the validation sample are expected to have a greater influence.

Building on this intuition, we define a pseudo-labeling approach to evaluate the effectiveness of influence estimators. Given a validation sample $z_{\text{val}} = (x_{\text{val}}, y_{\text{val}})$, the pseudo-label for a training sample $z_{\text{tr}} = (x_{\text{tr}}, y_{\text{tr}})$ is defined as:

$$\tilde{y}_{\text{val,tr}} = \begin{cases} 1, & \text{if} \quad y_{\text{tr}} = y_{\text{val}}, \\ 0, & \text{otherwise.} \end{cases} \quad (23)$$

Therefore, for each validation sample $z_{\text{val}}$, we compute the influence of all training samples, $\{\mathcal{I}(z_n, z_{\text{val}})\}_{n=1}^N$, and evaluate the performance of the estimated influence using ROC AUC and Recall metrics. For Recall, we compute the percentage of training points with the same class as the valida-

*Table 5.* ROC AUC (%) and Recall (%) for identifying influential samples in image generation tasks across different influence estimation methods. **Bold** indicates the best results and underline denotes the second-best results. The results for TracIn and DataInf were generated by executing the official implementation provided by DataInf. [†] denotes results reported in the original DataInf paper.

| Task | Method | ROC AUC | Recall |
|---|---|---|---|
| Style Generation | TracIn | 60.48±6.77 | 49.00±8.13 |
| | DataInf | 61.13±7.10 | 51.84±6.71 |
| | TracIn[†] | 69.2±0.7 | 53.3±0.8 |
| | DataInf[†] | 82.0±0.5 | 68.7±0.6 |
| | VM | **92.19±5.25** | **79.85±8.62** |
| | FVM | 92.03±5.27 | 79.62±8.61 |
| Subject Generation | TracIn | 62.95±25.53 | 19.61±23.73 |
| | DataInf | 57.73±23.90 | 30.39±26.03 |
| | TracIn[†] | 82.0±0.0 | 21.0±0.3 |
| | DataInf[†] | 86.5±0.0 | 31.5±0.3 |
| | VM | 92.69±11.14 | 52.94±31.95 |
| | FVM | **92.87±10.44** | **55.39±30.04** |

tion sample among the $s$ smallest influential training points. Here, $s$ is set to be the number of training examples per class. The mean and standard derivation of these metrics is then computed across all validation samples to summarize the results.

We compare our proposed method against TracIn (Pruthi et al., 2020) and DataInf (Kwon et al., 2024). The experiment is conducted by fine-tuning LoRA (Hu et al., 2022) on the Llama-2-13B-chat model (Touvron et al., 2023) for the text-to-text generation task. We evaluate two tasks in text generation task: (i) Sentence transformations, and (ii) Math word problems (with reasoning). The detailed description for each task and dataset is given in Appendix B.2.

The results are presented in Table 4. Our proposed methods, VM and FVM consistently achieve the highest performance across both tasks. Specifically, we note that FVM does not always achieve the highest scores. This can be attributed to the fact that our fine-tuning process employed a simplified version of the flat minima solver, LPF (Bisla et al., 2022), using $M = 1$ instead of the recommended $M = 8$ in the original work. This suboptimal choice of hyperparameters may have limited the full potential of FVM, preventing it from consistently outperforming all baselines. Nonetheless, the overall trend still demonstrates the robustness and effectiveness of our method in identifying influential samples across text generation tasks.

### 4.4. Influential Sample Identification in Image Generation

Following Kwon et al. (2024), we further evaluate our proposed influence estimation method for identifying influ-

*Table 6.* Ablation study on CIFAR-10N dataset under the "worst" setting. The results are evaluated using ROC AUC and AP. Components included are marked with a ✓. **Bold** indicates the best results.

| Loss change standard | ours | Parameter change standard | ours | ROC AUC | AP |
|---|---|---|---|---|---|
| ✓ | | ✓ | | 41.52 | 55.40 |
| ✓ | | | ✓ | 33.38 | 47.64 |
| | ✓ | ✓ | | 95.93 | 91.38 |
| | ✓ | | ✓ | **96.46** | **94.97** |

ential samples in image generation tasks, using the same experimental settings as described in Appendix B.2. We evaluate two tasks in text-to-image generation: (i) style generation and (ii) subject generation. For the style generation task, we combine three publicly available image-text pair datasets, each representing a different style: cartoons (Norod78, 2023), pixel-art (Jainr3, 2023), and line sketches (Zoheb, 2023). Further details of the experimental setup are provided in Appendix B.3.

The results are presented in Table 5. Our proposed methods, VM and FVM consistently achieve the best performance across both tasks. Similar to the result in Table 4, FVM does not always achieve the highest scores, with aligns with our earlier discussion in Section 4.3.

### 4.5. Ablation Study

We begin with an ablation study to evaluate the effectiveness of the two key components of our proposed Influence Function: loss change and parameter change. For loss change, we compare the term $g_{z_{\mathrm{val}}}^\top \Delta\theta$ used in the standard IF, referred to as "standard," with $\Delta\theta^\top \tilde{H}_{\mathrm{val}} \Delta\theta$ used in our proposed IF, referred to as "ours." For parameter change, we compare $-\epsilon H_{\mathrm{tr}}^{-1} g_{z_{\mathrm{tr}}}$ as "standard" with $-\epsilon \tilde{H}_{\mathrm{val}}^{-1} \tilde{g}_{z_{\mathrm{tr}}}$ as "ours." We conduct experiments on CIFAR-10N dataset under the "worst" setting. The model is tuned on the validation set with SAM (Foret et al., 2021) used as the flat minima solver, and the experimental setup is consistent with the details provided in Section 4.1.

The results are presented in Table 6. As observed, our proposed loss change term significantly enhances influence estimation performance, which supports our analysis highlighting the importance of the second-order approximation. The proposed parameter change term actually reduces performance when combined with the standard loss change term. However, when applied alongside our proposed loss change, it leads to a performance improvement.

## 5. Conclusion

In this work, we revisited the Influence Function (IF) and identified a fundamental limitation when applied to deep

neural networks, particularly in the presence of noisy training data. Through theoretical analysis and empirical observations, we demonstrated that existing IF methods suffer due to the sharpness of the validation risk, which leads to unreliable influence estimates. To address this issue, we established a novel connection between influence estimation error, validation set risk, and its sharpness, highlighting the importance of flat validation minima for accurate influence estimation. Building on this insight, we employ a second-order approximation to minimize the impact of vanishing gradients and introduce a refined parameter change estimation method tailored for flat test minima. Extensive experiments across various tasks demonstrated the superior performance of our approach, validating its effectiveness in enhancing influence estimation accuracy.

## Acknowledgments

This work was supported by AI for Science Foundation of Fudan University (FudanX24AI028), the National Natural Science Foundation of China (Grant No. 62472097), the Hong Kong Research Grants Council (Project No. 22303424), and GuangDong Basic and Applied Basic Research Foundation (Grant No. 2025A1515010259).

## Impact Statement

This paper presents work whose goal is to advance the field of Machine Learning, specifically by enhancing influence estimation in deep neural networks. Our approach improves the reliability of influence functions, which benefits applications such as dataset pruning, model debugging, and fairness analysis. By helping detect mislabeled or anomalous samples, our methods can lead to more robust and trustworthy models. However, we note that careful interpretation is necessary to prevent potential biases or unintended consequences when applying influence-based techniques in sensitive domains.

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

# A. Implementation Details

---

**Algorithm 1** Influence Function in Flat Validation Minima

---

1: **Input:** training set $S_{\text{tr}}$, validation set $S_{\text{val}}$, parameter pre-trained on training set $\theta^\star$, training sample $z_{\text{tr}}$
2:
3: # Solving for flat validation minima
4: Initialize $\theta \leftarrow \theta^\star$
5: $\tilde{\theta} \leftarrow$ Solving $\hat{R}_{\text{val}}^\gamma(\theta)$ (e.g. Sharpness-Aware Optimization)
6:
7: # Computing influence
8: $\mathcal{I}(z_{\text{tr}}, S_{\text{val}}) \leftarrow \tilde{g}_{z_{\text{tr}}}^\top \tilde{H}_{\text{val}}^{-1} \tilde{g}_{z_{\text{tr}}}$
9:
10: **Output:** $\mathcal{I}(z_{\text{tr}}, S_{\text{val}})$

---

Algorithm 1 presents the algorithm for computing the influence of a single training sample $z_{\text{tr}}$ on the validation set $S_{\text{val}}$. The algorithm first optimizes for the flat validation minima before computing the influence function.

To efficiently approximate the inverse Hessian matrix, we adopt a diagonal preconditioner, defined as: $\tilde{P}_{\text{val}} := \texttt{diag}(\tilde{F}_{\text{val}})$, where

$$\tilde{F}_{\text{val}} = \frac{1}{M} \sum_{m=1}^{M} \tilde{g}_{z_m} \tilde{g}_{z_m}^\top \tag{24}$$

represents the empirical Fisher Information matrix computed on the validation set $S_{\text{val}}$, serving as an approximation of Hessian. Assuming that the diagonal elements dominate the empirical Fisher Information matrix, we approximate $\tilde{H}_{\text{val}}$ by $\tilde{P}_{\text{val}}$ using $\mathcal{I}(z_{\text{tr}}, S_{\text{val}})$, resulting in:

$$\mathcal{I}(z_{\text{tr}}, S_{\text{val}}) \approx -\tilde{g}_{z_{\text{tr}}}^\top \tilde{P}_{\text{val}}^{-1} \tilde{g}_{z_{\text{tr}}}. \tag{25}$$

Here, $\tilde{P}_{\text{val}}^{-1}$ is the inverse of the diagonal matrix $\tilde{P}_{\text{val}}$ and can be easily computed due to its diagonal structure. Empirically, we find that this approximation significantly reduces computational overhead while maintaining accuracy in influence estimation.

# B. Experiment Details

## B.1. Mislabeled Sample Detection and Training Sample Relabeling

**Datasets**. Our evaluation is conducted on the CIFAR-10N and CIFAR-100N datasets (Wei et al., 2022), which are real-world noisy label variants of the CIFAR-10 and CIFAR-100 datasets (Krizhevsky et al., 2009). CIFAR-10N includes five noisy label settings: "Aggregate", "Random 1", "Random 2", "Random 3" and "Worst." CIFAR-100N provides a single noisy label setting, "Noisy." These noisy labels arise from human annotation errors. For our evaluation, we use the "Aggregate", "Random 1" and "Worst" settings for CIFAR-10N, and the "Noisy" setting for CIFAR-100N.

**Training**. We follow the experiment settings in Wei et al. (2022). The ResNet-34 (He et al., 2016) is utilized for both CIFAR-10N and CIFAR-100N. The basic hyper-parameters settings are listed as follows: minibatch size (128), optimizer (SGD), initial learning rate (0.1), momentum (0.9), weight decay (0.0005), number of epochs (100), and learning rate decay (0.1 at 50 epochs). Standard data augmentation, including random crop and random horizontal flip, is applied to each dataset.

**Hyper-Parameters**. The hyper-parameters for each influence estimation approach are outlined as follows.

For TracIn, we use five checkpoints evenly sampled from the training process, corresponding to epochs 20, 40, 60, 80, and 100.

For DataInf, the damping parameter is set as in the original paper: $\lambda_l = 0.1 \times (Nd_l)^{-1} \sum_{n=1}^{N} \nabla_{\theta_l} \ell_n^\top \nabla_{\theta_l} \ell_n$, where $l$ represents the corresponding model layer and $d_l$ denotes the number of parameters in that layer.

For LiSSA* and DataInf*, we select the checkpoint at epoch 60, which achieves the best AP performance among epochs 10 to 100, evaluated at intervals of 10 epochs.

For GEX, we follow the procedure outlined in the original paper, generating 32 Geometric Ensemble (GE) to obtain the

*Table 7.* Description of the sentence transformations task prompts and templates. We consider 10 different types of sentence transformations, following the setup of Kwon et al. (2024).

| Sentence transformations | Example transformation of *"Sunrises herald hopeful tomorrows"*: |
|---|---|
| Reverse Order of Words | tomorrows. hopeful herald Sunrises |
| Capitalize Every Other Letter | sUnRiSeS hErAlD hOpEfUl tOmOrRoWs. |
| Insert Number 1 Between Every Word | Sunrises 1herald 1hopeful 1tomorrows. |
| Replace Vowels with * | S*nr*s*s h*r*ld h*p*f*l t*m*rr*ws. |
| Double Every Consonant | SSunrriisseess hheralld hhopeffull ttomorrows. |
| Capitalize Every Word | Sunrises Herald Hopeful Tomorrows. |
| Remove All Vowels | Snrss hrld hpfl tmrrws. |
| Add 'ly' To End of Each Word | Sunrisesly heraldly hopefully tomorrows.ly |
| Remove All Consonants | uie ea oeu ooo. |
| Repeat Each Word Twice | Sunrises Sunrises herald herald hopeful hopeful tomorrows. tomorrows. |

final results. Each GE is generated by tuning the trained model on the validation set, with tuning settings consistent with those used for our VM and FVM, as described below.

For our proposed VM and FVM, to obtain $\tilde{\theta}$, we tune the trained model on the validation set with the following hyperparameter settings: minibatch size (128), optimizer (SGD), initial learning rate (0.01), momentum (0.9), weight decay (0.0005), number of steps (1000), and learning rate decay (cosine). For FVM, SAM (Foret et al., 2021) is used as the flat minima solver, with the hyperparameter $\gamma$ set to 0.05 for CIFAR-10N and 0.1 for CIFAR-100N, in accordance with the original paper. In this experiment, we treat the samples with large influence scores as detected noisy samples.

### B.2. Text-To-Text Generation

**Datasets**. Following Kwon et al. (2024), we consider three text-to-text generation tasks: sentence transformations and math problems (w/ reasoning).

- Sentence transformations: The model is tasked with applying a specific transformation to a given sentence. Further details can be found in Table 7.

- Math problems (w/ reasoning): The model is presented with an arithmetic word problem and is expected to provide an intermediate reasoning step and a direct numerical answer before arriving at the final answer. A detailed description of the prompts and templates is provided in Table 8.

Each dataset contains 10 distinct classes, with 100 total data points in each class. We partitioned the 100 examples into 90 training data points (used for LoRA) and 10 validation data points for influence estimation.

**Training**. We use LoRA (Hu et al., 2022) to Llama-2-13B-chat (Touvron et al., 2023). We apply LoRA to every query and value matrix of the attention layer in the Llama-2-13B-chat model. The basic hyper-parameters settings are listed as follows: minibatch size (64), optimizer (AdamW (Loshchilov & Hutter, 2019)), initial learning rate ($3 \times 10^{-4}$), number of epochs (10), and learning rate decay (cosine). For LoRA hyperparameters, we set $r = 8$ and $\alpha = 32$. The training was performed using the HuggingFace Peft library (Mangrulkar et al., 2022)

**Loss**. We used a negative log-likelihood of a generated response as a loss function. For a sequence of input tokens $x = (x_1, \cdots, x_{T_1})$ and the corresponding sequence of target tokens $y = (y_1, \cdots, y_{T_2})$, suppose the Llama-2-13B generates a sequence of output tokens $f_\theta(x) = (f_\theta(x)_1, \cdots, f_\theta(x)_{T_2})$. $f_\theta(x)$ has the same size of $T_2$ and is generated in an auto-regressive manner. We set $T_1 = T_2 = 512$. Then, the loss function is $\ell(y, f_\theta(x)) = -\sum_{t=1}^{T_2} \log p(y_t \mid f_\theta(x)_1, \cdots, f_\theta(x)_{t-1})$.

**Hyper-Parameters**. The hyper-parameters for each influence estimation approach are outlined as follows.

For TracIn, we use the final checkpoint for influence computation, following Kwon et al. (2024).

For DataInf, the damping parameter is set as in the original paper: $\lambda_l = 0.1 \times (Nd_l)^{-1} \sum_{n=1}^{N} \nabla_{\theta_l} \ell_n^\top \nabla_{\theta_l} \ell_n$, where $l$ represents the corresponding model layer and $d_l$ denotes the number of parameters in that layer.

*Table 8.* Description of the math problem task templates. We consider 10 different types of math word problems, following the setup of Kwon et al. (2024).

| Math word problems | Template prompt question |
|---|---|
| Remaining pizza slices | Lisa ate A slices of pizza and her brother ate B slices from a pizza that originally had C slices. How many slices of the pizza are left?
*Reason:* Combined slices eaten = A + B. Left = C - (A + B). |
| Chaperones needed for trip | For every A students going on a field trip, there are B adults needed as chaperones. If C students are attending, how many adults are needed?
*Reason:* Adults needed = (B * C) // A. |
| Total number after purchase | In an aquarium, there are A sharks and B dolphins. If they bought C more sharks, how many sharks would be there in total?
*Reason:* Total sharks = A + C. |
| Total game points | Michael scored A points in the first game, B points in the second, C in the third, and D in the fourth game. What is his total points?
*Reason:* Total points = A + B + C + D. |
| Total reading hours | Emily reads for A hours each day. How many hours does she read in total in B days?
*Reason:* Total hours read = A * B. |
| Shirt cost after discount | A shirt costs A. There's a B-dollar off sale. How much does the shirt cost after the discount?
*Reason:* Cost after discount = A - B. |
| Area of a garden | A rectangular garden has a length of A meters and a width of B meters. What is its area?
*Reason:* Area = A * B. |
| Total savings | If Jake saves A each week, how much will he save after B weeks?
*Reason:* Total savings = A * B. |
| Number of cupcake boxes | A bakery sells cupcakes in boxes of A. If they have B cupcakes, how many boxes can they fill?
*Reason:* Boxes filled = B // A. |
| Interest earned | John invests A at an annual interest rate of B%. How much interest will he earn after C years?
*Reason:* Interest = (A * B * C) // 100. |

For our proposed VM and FVM, to obtain $\tilde{\theta}$, we further fine-tune the previously tuned LoRA model (i.e., the model obtained after the initial fine-tuning on the training set) on the validation set for 5 epochs with an initial learning rate of $3 \times 10^{-5}$. For FVM, we set $\epsilon = 0.01$. LPF (Bisla et al., 2022) is employed as the flat minima solver, with the hyper-parameters $M = 1$ and $\gamma = 0.001$ for all tasks, in accordance with the original paper.

### B.3. Text-To-Image Generation

**Datasets**. Here, we briefly describe the datasets used in image generation tasks.

For style generation, we fine-tune the model by appending a style description to the text prompt, instructing it to generate images in a specific style. We use 200 training image-text pairs and 50 validation image-text pairs, resulting in a total of 600 training data points and 150 validation data points. We use the DreamBooth dataset (Ruiz et al., 2023), which contains 30 different subjects The text follows the format: "Generate an image in a specific custom style. text-data", where custom is replaced with either "cartoon", "pixelart", or "black and white line sketch", and text-data is replaced with a text sequence from the training dataset. A detailed description of the style prompts is provided in Table 9.

For subject generation, we use Google's DreamBooth dataset (Ruiz et al., 2023), which contains 31 unique subjects across categories such as backpack, dog, bowl, and sneaker. Each subject has four to six images, with three examples per subject used for training and the remaining for validation. For each subject, 3 data points are used for the training dataset and 1 to 3 data points are used for the validation dataset. To enable the model to distinguish between different subjects, we assign a unique random string to each subject in the prompt. For instance, two different dogs are referenced as "a i0VpE dog" and "a oHwLM dog" in the training and validation sets, respectively.

**Training**. We use LoRA (Hu et al., 2022) to fine-tune stable-diffusion-v1.5 (Rombach et al., 2022). We apply LoRA to

*Table 9.* Description of the text-to-image generation task templates. Each style has 200 training image-text pairs and 150 validation image-text pairs, following the setup of Kwon et al. (2024).

| Image style | Text prompt |
|---|---|
| Cartoon | Generate an image in a specific cartoon style. A text sequence of the original dataset which describes an image. |
| Pixel Art | Generate an image in a specific pixelart style. A text sequence of the original dataset which describes an image. |
| Sketch | Generate an image in a specific black and white line sketch style. A text sequence of the original dataset which describes an image. |

*Table 10.* Area Under the Receiver Operating Characteristic Curve (ROC AUC) (%) and Average Precision (AP) (%) for identifying mislabeled samples on the CIFAR-10N/-100N datasets across different influence estimation methods. Results (mean±std) are reported over 3 random runs. **Bold** indicates the best results and underline denotes the second-best results.

| Method | CIFAR-10N | | | | | | CIFAR-100N | |
|---|---|---|---|---|---|---|---|---|
| | Aggre | | Random | | Worst | | Noisy | |
| | ROC AUC | AP | ROC AUC | AP | ROC AUC | AP | ROC AUC | AP |
| EK-FAC | 57.03±0.55 | 39.71±1.41 | 69.85±1.77 | 57.70±1.66 | 72.18±9.20 | 72.81±8.64 | 60.30±1.30 | 59.37±0.87 |
| EK-FAC* | 70.02±1.40 | 60.51±1.42 | 81.55±0.49 | 76.58±0.07 | 80.24±5.13 | 81.72±4.66 | 62.72±1.51 | 64.86±1.02 |
| VM | 95.18±0.15 | 76.31±0.21 | 95.92±0.10 | 87.35±0.45 | 95.88±0.13 | 94.27±0.21 | 89.77±0.08 | 83.81±0.18 |
| FVM | **96.14±0.06** | **79.53±0.27** | **96.63±0.03** | **88.82±0.26** | **96.46±0.08** | **94.97±0.12** | **90.80±0.04** | **85.41±0.05** |

every attention layer in the stable-diffusion-v1.5 model. The basic hyper-parameters settings are listed as follows: minibatch size (4), optimizer (AdamW (Loshchilov & Hutter, 2019)), initial learning rate ($10^{-4}$), number of steps (10000), and learning rate decay (cosine). For LoRA hyperparameters, we set $r = 2$ and $\alpha = 2$. The training was performed using the HuggingFace Peft library (Mangrulkar et al., 2022).

**Loss**. We use a negative log-likelihood of a generated image as a loss function. For a sequence of input tokens $x = (x_1, \cdots, x_T)$ and the corresponding target image $y$, we compute a negative log-likelihood $\ell(y, f_\theta(x)) = -\log p(y \mid f_\theta(x))$. We set $T = 77$.

**Hyper-parameters**. The hyper-parameter settings for image generation tasks are the same as those in text generation tasks.

## C. Addtional Experiment Results

### C.1. Comparison with EK-FAC

We compare our proposed VM/FVM approach with EK-FAC (Grosse et al., 2023) on mislabeled sample detection tasks, following the experimental setup described in Appendix B.1. The results are presented in Table 10, where * indicates performance based on the best training checkpoints. As shown, our method consistently outperforms the EK-FAC-based influence function in identifying mislabeled data.

### C.2. Validation Set Size

To investigate the impact of the validation set size on our approach further, we conducted experiments comparing different IF approaches for mislabel detection on the CIFAR-10N Worst dataset with varying validation set sizes. The results (ROC AUC/AP) are reported in Table 11, with the performance drop relative to the size of 10,000 shown in parentheses. As observed, the performance of all approaches tends to degrade. Nevertheless, our method maintains strong performance despite the reduced data size and still outperforms other approaches by a significant margin.

*Table 11.* Area Under the Receiver Operating Characteristic Curve (ROC AUC) (%) and Average Precision (AP) (%) for identifying mislabeled samples on the CIFAR-10N Worst datasets across different influence estimation methods. Results (mean±std) are reported over 3 random runs. Performance changes (%) are computed relative to the results under the 10,000-sample validation set.

| Metric | Size | LiSSA | VM | FVM |
|---|---|---|---|---|
| ROC AUC | 10000 | $65.75 \pm 0.39$ | $95.88 \pm 0.13$ | $96.46 \pm 0.08$ |
| | 5000 | $65.84 \pm 0.35$ ($\uparrow 0.13$) | $95.31 \pm 0.19$ ($\downarrow 0.59$) | $95.68 \pm 0.23$ ($\downarrow 0.80$) |
| | 2000 | $65.60 \pm 0.49$ ($\downarrow 0.22$) | $94.37 \pm 0.27$ ($\downarrow 1.57$) | $94.86 \pm 0.21$ ($\downarrow 1.65$) |
| | 1000 | $64.79 \pm 0.94$ ($\downarrow 1.46$) | $93.07 \pm 0.65$ ($\downarrow 2.93$) | $93.72 \pm 0.53$ ($\downarrow 2.84$) |
| AP | 10000 | $68.35 \pm 0.74$ | $94.27 \pm 0.21$ | $94.97 \pm 0.12$ |
| | 5000 | $68.35 \pm 0.95$ ($\downarrow 0.00$) | $93.44 \pm 0.14$ ($\downarrow 0.88$) | $93.71 \pm 0.22$ ($\downarrow 1.32$) |
| | 2000 | $67.75 \pm 1.27$ ($\downarrow 0.87$) | $92.11 \pm 0.16$ ($\downarrow 2.29$) | $92.60 \pm 0.14$ ($\downarrow 2.49$) |
| | 1000 | $66.54 \pm 1.74$ ($\downarrow 2.64$) | $90.26 \pm 0.69$ ($\downarrow 4.25$) | $90.94 \pm 0.54$ ($\downarrow 4.24$) |

*Table 12.* Area Under the Receiver Operating Characteristic Curve (ROC AUC) (%) and Average Precision (AP) (%) for identifying mislabeled samples on the CIFAR-10N Worst and CIFAR-100N Noisy datasets across different sharpness-aware optimizers equipped to FVM. Results (mean±std) are reported over 3 random runs. **Bold** indicates the best results and underline denotes the second-best results.

| Method | CIFAR-10N Worst | | CIFAR-100N Noisy | |
|---|---|---|---|---|
| | ROC AUC | AP | ROC AUC | AP |
| SAM | $96.46 \pm 0.08$ | $94.97 \pm 0.12$ | $90.80 \pm 0.04$ | $85.41 \pm 0.05$ |
| ASAM | $96.71 \pm 0.14$ | $95.11 \pm 0.19$ | $90.83 \pm 0.11$ | $85.25 \pm 0.22$ |
| F-SAM | **$96.76 \pm 0.10$** | **$95.29 \pm 0.14$** | **$91.25 \pm 0.04$** | **$86.06 \pm 0.03$** |

## C.3. Sharpness-Aware Optimizer

We compare three different sharpness-aware optimizer, including SAM (utilized in our initial approach), ASAM (Kwon et al., 2021), and F-SAM (Li et al., 2024), on CIFAR-10N Worst and CIFAR-100N Noisy datasets. The results (ROC AUC) are presented in Table 12. As observed, different sharpness-aware optimizers do affect the final results, with F-SAM achieving the best performance. This supports our hypothesis that better flat validation minima can lead to more accurate influence function estimation.

## C.4. Inverse Hessian Approximation

To quantitatively discover the impact of the approximation on the inverse Hessian, we conduct experiments on the mislabel detection task. Specifically, we replace the computation of the inverse Hessian $\tilde{H}_{val}^{-1}$ in Equation (21) from diagonal Fisher to LiSSA (Koh & Liang, 2017) and DataInf (Kwon et al., 2024).

Note that the accelerated approximations in LiSSA and DataInf both rely on the inverse Hessian-vector product (iHVP). However, the initial product $\tilde{H}_{val}^{-1}\tilde{g}_{z_{tr}}$ in Equation (21) depends on the specific training sample $z_{tr}$, which requires recomputation for each training sample. As a result, we cannot directly apply these methods to our proposed influence function. To address this, we introduce a random vector $V \in R^{|\theta| \times 1}$, where each element is sampled from a standard normal distribution, i.e., $V_i \sim \mathcal{N}(0, 1)$. With this, Equation (21) becomes $\tilde{g}_{z_{tr}}^\top \tilde{H}_{val}^{-1} V V^\top \tilde{g}_{z_{tr}}$, and we have $\mathbb{E}[\tilde{g}_{z_{tr}}^\top \tilde{H}_{val}^{-1} V V^\top \tilde{g}_{z_{tr}}] = \tilde{g}_{z_{tr}}^\top \tilde{H}_{val}^{-1} \tilde{g}_{z_{tr}}$. By using the random vector $V$, we can directly apply the iHVP trick to $\tilde{H}_{val}^{-1} V$ and compute the inverse Hessian based on LiSSA or DataInf. In practice, we sample 5 different $V$ to ensure stability and reduce the variance of the approximation.

The results (ROC AUC/AP) are reported in Table 13. Theoretically, from the perspective of computational complexity and estimation fidelity, LiSSA and DataInf are expected to provide more accurate approximations of the inverse Hessian than the diagonal approximation. However, in our current experiments, we observe that the performance of the diagonal approximation is competitive with, and in some cases even superior to, LiSSA and DataInf.

One possible reason is that LiSSA and DataInf involve more sensitive hyperparameters (e.g., number of iterations), and tuning them appropriately for each setting is non-trivial. Despite this, the diagonal approximation, which is significantly

*Table 13.* Area Under the Receiver Operating Characteristic Curve (ROC AUC) (%) and Average Precision (AP) (%) for identifying mislabeled samples on the CIFAR-10N/-100N datasets using different inverse Hessian approximation methods. **Bold** indicates the best results and underline denotes the second-best results.

| Method | Inverse Hessian Approximation | CIFAR-10N | | | | | | CIFAR-100N | |
| | | Aggre | | Random | | Worst | | Noisy | |
| | | ROC AUC | AP | ROC AUC | AP | ROC AUC | AP | ROC AUC | AP |
|---|---|---|---|---|---|---|---|---|---|
| VM | LiSSA | 95.22 | 67.36 | 95.67 | 78.90 | 95.00 | 89.26 | 88.88 | 80.89 |
| | DataInf | **95.37** | 72.09 | **95.98** | 83.26 | 95.57 | 91.09 | 89.58 | 82.81 |
| | Diagonal (ours) | 95.18 | **76.31** | 95.92 | **87.35** | **95.88** | **94.27** | **89.77** | **83.81** |
| FVM | LiSSA | 95.95 | 70.71 | 96.17 | 80.16 | 95.33 | 89.56 | 89.48 | 81.81 |
| | DataInf | 96.18 | 76.15 | 96.58 | 84.85 | 96.02 | 91.54 | 90.24 | 83.81 |
| | Diagonal (ours) | **96.63** | **88.82** | **96.63** | **88.82** | **96.46** | **94.97** | **90.80** | **85.41** |

more efficient, achieves consistently strong performance across all datasets. This suggests that our use of the diagonal approximation, while simplistic, does not lead to a substantial degradation in influence estimation accuracy in practice.

## D. Proofs

### D.1. Proof of Theorem 3.2

*Proof.* Following Definition 3.1, we have

$$\mathcal{E}(\mathcal{I}) = \mathbb{E}_{z \sim \mathcal{D}} \left[ 1 \left\{ \text{sgn} \left( \mathcal{I}(z, S_{\text{val}}) \right) \neq \text{sgn} \left( \mathcal{I}^\star(z, S_{\text{val}}) \right) \right\} \right] \tag{26}$$

$$= \mathbb{P}_{z \sim \mathcal{D}} \left[ \text{sgn} \left( \mathcal{I}(z, S_{\text{val}}) \right) \neq \text{sgn} \left( \mathcal{I}^\star(z, S_{\text{val}}) \right) \right]. \tag{27}$$

Next, we decompose this probability as:

$$\mathcal{E}(\mathcal{I}) = \mathbb{P}_{z \sim \mathcal{D}} \left[ \text{sgn} \left( \mathcal{I}^\star(z, S_{\text{val}}) \right) = 1 \cap \mathcal{I}(z, S_{\text{val}}) < 0 \right] \tag{28}$$

$$+ \mathbb{P}_{z \sim \mathcal{D}} \left[ \text{sgn} \left( \mathcal{I}^\star(z, S_{\text{val}}) \right) = -1 \cap \mathcal{I}(z, S_{\text{val}}) > 0 \right]. \tag{29}$$

Let $p = \mathbb{P}_{z \sim \mathcal{D}} \left[ \text{sgn} \left( \mathcal{I}^\star(z, S_{\text{val}}) \right) = 1 \right]$, we can express $\mathcal{E}(\mathcal{I})$ as:

$$\mathcal{E}(\mathcal{I}) = p \, \mathbb{P}_{z \sim \mathcal{D}} \left[ \mathcal{I}(z, S_{\text{val}}) < 0 \mid \text{sgn} \left( \mathcal{I}^\star(z, S_{\text{val}}) \right) = 1 \right] \tag{30}$$

$$+ (1 - p) \, \mathbb{P}_{z \sim \mathcal{D}} \left[ \mathcal{I}(z, S_{\text{val}}) > 0 \mid \text{sgn} \left( \mathcal{I}^\star(z, S_{\text{val}}) \right) = -1 \right] \tag{31}$$

$$= p \, \mathbb{P}_{z \sim \mathcal{D}_+} \left[ \mathcal{I}(z, S_{\text{val}}) < 0 \right] + (1 - p) \, \mathbb{P}_{z \sim \mathcal{D}_-} \left[ \mathcal{I}(z, S_{\text{val}}) > 0 \right] \tag{32}$$

$$= p \mathbb{P}_{z \sim \mathcal{D}_+} \left[ \mathcal{I}(z, S_{\text{val}}) - \mathbb{E}_{z \sim \mathcal{D}_+} \left[ \mathcal{I}(z, S_{\text{val}}) \right] < -\mathbb{E}_{z \sim \mathcal{D}_+} \left[ \mathcal{I}(z, S_{\text{val}}) \right] \right] \tag{33}$$

$$+ (1 - p) \mathbb{P}_{z \sim \mathcal{D}_-} \left[ \mathcal{I}(z, S_{\text{val}}) - \mathbb{E}_{z \sim \mathcal{D}_-} \left[ \mathcal{I}(z, S_{\text{val}}) \right] > -\mathbb{E}_{z \sim \mathcal{D}_-} \left[ \mathcal{I}(z, S_{\text{val}}) \right] \right]. \tag{34}$$

Given the assumption that $\mathbb{E}_{z \sim \mathcal{D}_+} [\mathcal{I}(z, S_{\text{val}})] > 0$ and $\mathbb{E}_{z \sim \mathcal{D}_-} [\mathcal{I}(z, S_{\text{val}})] < 0$, by Hoeffding's inequality, we can bound these probabilities:

$$\mathcal{E}(\mathcal{I}) \leq p \exp \left( -\frac{2 \, \mathbb{E}_{z \sim \mathcal{D}_+} \left[ \mathcal{I}(z, S_{\text{val}}) \right]^2}{\left( \sup_{z \sim \mathcal{D}_+} \mathcal{I}(z, S_{\text{val}}) - \inf_{z \sim \mathcal{D}_+} \mathcal{I}(z, S_{\text{val}}) \right)^2} \right) \tag{35}$$

$$+ (1 - p) \exp \left( -\frac{2 \, \mathbb{E}_{z \sim \mathcal{D}_-} \left[ \mathcal{I}(z, S_{\text{val}}) \right]^2}{\left( \sup_{z \sim \mathcal{D}_+} \mathcal{I}(z, S_{\text{val}}) - \inf_{z \sim \mathcal{D}_-} \mathcal{I}(z, S_{\text{val}}) \right)^2} \right) \tag{36}$$

$$\leq \exp \left( -\frac{2\mu^2}{\left( \sup_{z \sim \mathcal{D}} \mathcal{I}(z, S_{\text{val}}) - \inf_{z \sim \mathcal{D}} \mathcal{I}(z, S_{\text{val}}) \right)^2} \right), \tag{37}$$

where $\mu = \min(|\mathbb{E}_{z \sim \mathcal{D}_+}[\mathcal{I}(z, S_{\text{val}})]|, |\mathbb{E}_{z \sim \mathcal{D}_-}[\mathcal{I}(z, S_{\text{val}})]|)$

Now, recall the definition of Influence Function, where $\mathcal{I}(z, S_{\text{val}}) = \hat{R}_{\text{val}}(\theta_z) - \hat{R}_{\text{val}}(\theta)$. Substituting this, we can rewrite the bound as:

$$\mathcal{E}(\mathcal{I}) \leq \exp\left(-\frac{2\mu^2}{\left(\sup_{z \sim \mathcal{D}} \hat{R}_{\text{val}}(\theta_z) - \inf_{z \sim \mathcal{D}} \hat{R}_{\text{val}}(\theta_z)\right)^2}\right), \tag{38}$$

and $\mu = \min\left(|\mathbb{E}_{z \sim \mathcal{D}_+}\left[\hat{R}_{\text{val}}(\theta_z)\right] - \hat{R}_{\text{val}}(\theta)|, |\mathbb{E}_{z \sim \mathcal{D}_-}\left[\hat{R}_{\text{val}}(\theta_z)\right] - \hat{R}_{\text{val}}(\theta)|\right)$.

Finally, considering that we assume a linear or second-order approximation with respect to the parameter change is valid, the perturbed parameter $\theta_z$ estimated by the Influence Function lies in the neighborhood of $\theta$. Thus, we have:

$$\sup_{z \sim \mathcal{D}} \hat{R}_{\text{val}}(\theta_z) \leq \max_{\|\Delta\| \leq \gamma} \hat{R}_{\text{val}}(\theta + \Delta), \tag{39}$$

where $\Delta$ is the parameter change and $\gamma = \sup_{z \sim D} \|\theta_z - \theta\|$.

Define $\hat{R}_{\text{val}}^{\gamma}(\theta) := \max_{\|\Delta\| \leq \gamma} \hat{R}_{\text{val}}(\theta + \Delta)$, the error bound becomes:

$$\mathcal{E}(\mathcal{I}) \leq \exp\left(-\frac{2\mu^2}{\left(\hat{R}_{\text{val}}^{\gamma}(\theta) - \inf_{z \sim \mathcal{D}} \hat{R}_{\text{val}}(\theta_z)\right)^2}\right) \leq \exp\left(-\frac{2\mu^2}{\hat{R}_{\text{val}}^{\gamma}(\theta)^2}\right). \tag{40}$$

This completes the proof. $\qquad\square$

### D.2. Proof of Corollary 3.3

*Proof.* From the definition of $\hat{\mathcal{E}}_S(\mathcal{I})$, we have:

$$\hat{\mathcal{E}}_S(\mathcal{I}) = \frac{1}{N} \sum_{n=1}^{N} [1\{\text{sgn}(\mathcal{I}(z_n, S_{\text{val}})) \neq \text{sgn}(\mathcal{I}^\star(z_n, S_{\text{val}}))\}]. \tag{41}$$

Let $X_{z_n} = 1\{\text{sgn}(\mathcal{I}(z_n, S_{\text{val}})) \neq \text{sgn}(\mathcal{I}^\star(z_n, S_{\text{val}}))\}$. Applying Hoeffding's inequality, we have:

$$\mathbb{P}\left[\frac{1}{N} \sum_{n=1}^{N} X_{z_n} - \mathbb{E}_{z \sim \mathcal{D}}[X_z] \geq t\right] \leq \exp\left(-2Nt^2\right), \tag{42}$$

where $t > 0$. Recall the definition of $R(\mathcal{I})$, this inequality can be rewritten as:

$$\mathbb{P}\left[\hat{\mathcal{E}}_S(\mathcal{I}) - \mathcal{E}(\mathcal{I}) \geq t\right] \leq \exp\left(-2Nt^2\right). \tag{43}$$

Let $\delta = \exp\left(-2Nt^2\right)$. Solving for $t$, we have $t = \sqrt{-\frac{\log \delta}{2N}}$. Substituting $t$ back into the inequality gives:

$$\mathbb{P}\left[\hat{\mathcal{E}}_S(\mathcal{I}) - \mathcal{E}(\mathcal{I}) \geq \sqrt{-\frac{\log \delta}{2N}}\right] \leq \delta. \tag{44}$$

Thus, with probability at least $1 - \delta$, the following inequality holds:

$$\hat{\mathcal{E}}_S(\mathcal{I}) - \mathcal{E}(\mathcal{I}) < \sqrt{-\frac{\log \delta}{2N}}. \tag{45}$$

Finally, using Theorem 3.2, we have

$$\hat{\mathcal{E}}_S(\mathcal{I}) < \mathcal{E}(\mathcal{I}) + \sqrt{-\frac{\log \delta}{2N}} \leq \exp\left(-\frac{2\mu^2}{\hat{R}_{\text{val}}^{\gamma}(\theta)^2}\right) + \sqrt{-\frac{\log \delta}{2N}}. \tag{46}$$

This completes the proof. $\qquad\square$

# E. Derivations

## E.1. The Parameter Change $\tilde{\theta}_{z_{tr}} - \tilde{\theta}$

Here, for completeness, we derive the parameter change $\tilde{\theta}_{z_{tr}} - \tilde{\theta}$ in the context of loss minimization. This derivation builds upon the analysis of $\theta^{\star}_{z_{tr}} - \theta^{\star}$ presented in Koh & Liang (2017).

Recall that $\tilde{\theta}$ minimizes the following SAM object:

$$\hat{R}^{\gamma}_{\text{val}}(\theta) := \max_{\|\Delta\| \leq \gamma} \hat{R}_{\text{val}}(\theta + \Delta) = \max_{\|\Delta\| \leq \gamma} \frac{1}{M} \sum_{m=1}^{M} \ell(z_m, \theta + \Delta). \tag{47}$$

We further assume that $R$ is twice-differentiable and strongly convex in $\theta$, i.e.,

$$\tilde{H}_{\text{val}} := \nabla^2_{\theta} \hat{R}_{\text{val}}(\tilde{\theta}) = \frac{1}{M} \sum_{m=1}^{M} \nabla^2_{\tilde{\theta}} \ell(z_m, \tilde{\theta}), \tag{48}$$

exists and is positive definite. This guarantees the existence of $\tilde{H}^{-1}_{\text{val}}$, which we will use in the subsequent derivation.

The perturbed parameters $\tilde{\theta}_{z_{tr}}$ can be written as

$$\tilde{\theta}_{z_{tr}} := \arg\min_{\theta} \max_{\|\Delta\| \leq \gamma} R_{\text{val}}(\theta + \Delta) + \epsilon \ell(z_{tr}, \theta + \Delta). \tag{49}$$

Since $\tilde{\theta}_{z_{tr}}$ is a minimizer of Equation (49), assuming $\gamma \to 0$, let us examine its first-order optimality conditions:

$$0 = \nabla_{\tilde{\theta}} R_{\text{val}}(\tilde{\theta}_{z_{tr}}) + \epsilon \nabla_{\tilde{\theta}} \ell(z, \tilde{\theta}_{z_{tr}}) \tag{50}$$

Next, since $\tilde{\theta}_{z_{tr}} \to \tilde{\theta}$ as $\epsilon \to 0$, we perform a Taylor expansion of the right-hand side:

$$0 \approx \left[ \nabla_{\tilde{\theta}} R_{\text{val}}(\tilde{\theta}) + \epsilon \nabla_{\tilde{\theta}} \ell(z_{tr}, \tilde{\theta}) \right] + \left[ \nabla^2_{\tilde{\theta}} R_{\text{val}}(\tilde{\theta}) + \epsilon \nabla^2_{\tilde{\theta}} \ell(z_{tr}, \tilde{\theta}) \right] \left( \tilde{\theta}_{z_{tr}} - \tilde{\theta} \right), \tag{51}$$

where we have dropped $o(\|\tilde{\theta}_{z_{tr}} - \tilde{\theta}\|)$ terms.

Sovling for $\tilde{\theta}_{z_{tr}} - \tilde{\theta}$, we get:

$$\tilde{\theta}_{z_{tr}} - \tilde{\theta} \approx - \left[ \nabla^2_{\tilde{\theta}} R_{\text{val}}(\tilde{\theta}) + \epsilon \nabla^2_{\tilde{\theta}} \ell(z_{tr}, \tilde{\theta}) \right]^{-1} \left[ \nabla_{\tilde{\theta}} R_{\text{val}}(\tilde{\theta}) + \epsilon \nabla_{\tilde{\theta}} \ell(z_{tr}, \tilde{\theta}) \right]. \tag{52}$$

Since $\tilde{\theta}$ is the minimizer of $\hat{R}^{\gamma}_{\text{val}}(\theta)$, we have $\nabla_{\tilde{\theta}} R_{\text{val}}(\tilde{\theta}) = 0$. Dropping $o(\epsilon)$ terms, then

$$\tilde{\theta}_z - \tilde{\theta} \approx -\epsilon \tilde{H}^{-1}_{\text{val}} \tilde{g}_{z_{tr}}, \tag{53}$$

where $\tilde{g}_{z_{tr}} = \nabla_{\tilde{\theta}} \ell(z_{tr}, \tilde{\theta})$.

## E.2. The Influence Function

Recall Equation (19), we have

$$\ell(z_{\text{val}}, \tilde{\theta}_{z_{tr}}) - \ell(z_{\text{val}}, \tilde{\theta}) \approx \tilde{g}^{\top}_{z_{\text{val}}} \Delta\theta + \frac{1}{2} \Delta\theta^{\top} \nabla^2_{\tilde{\theta}} \ell(z_{\text{val}}, \tilde{\theta}) \Delta\theta, \tag{54}$$

Combining with Equation (18), we have

$$\ell(z_{\text{val}}, \tilde{\theta}_{z_{tr}}) - \ell(z_{\text{val}}, \tilde{\theta}) \approx \tilde{g}^{\top}_{z_{\text{val}}} \left( -\epsilon \tilde{H}^{-1}_{\text{val}} \tilde{g}_{z_{tr}} \right) + \frac{1}{2} \left( -\epsilon \tilde{H}^{-1}_{\text{val}} \tilde{g}_{z_{tr}} \right)^{\top} \nabla^2_{\tilde{\theta}} \ell(z_{\text{val}}, \tilde{\theta}) \left( -\epsilon \tilde{H}^{-1}_{\text{val}} \tilde{g}_{z_{tr}} \right) \tag{55}$$

$$= -\epsilon \tilde{g}^{\top}_{z_{\text{val}}} \tilde{H}^{-1}_{\text{val}} \tilde{g}_{z_{tr}} + \frac{1}{2} \epsilon^2 \tilde{g}^{\top}_{z_{tr}} \tilde{H}^{-1}_{\text{val}} \nabla^2_{\tilde{\theta}} \ell(z_{\text{val}}, \tilde{\theta}) \tilde{H}^{-1}_{\text{val}} \tilde{g}_{z_{tr}}. \tag{56}$$

Since we want to measure the loss change with respect to removing the training sample $z_{\text{tr}}$, we define the Influence Function as follows:

$$\mathcal{I}(z_{\text{tr}}, z_{\text{val}}) := \tilde{g}_{z_{\text{val}}} \tilde{H}_{\text{val}}^{-1} \tilde{g}_{z_{\text{tr}}} + \frac{1}{2}\epsilon \tilde{g}_{z_{\text{tr}}}^{\top} \tilde{H}_{\text{val}}^{-1} \nabla_{\tilde{\theta}}^2 \ell(z_{\text{val}}, \tilde{\theta}) \tilde{H}_{\text{val}}^{-1} \tilde{g}_{z_{\text{tr}}}, \tag{57}$$

with $\epsilon > 0$. Next, we consider the influence on the validation set $S_{\text{val}}$, which can be defined as the sum of the influence on each validation sample:

$$\mathcal{I}(z_{\text{tr}}, S_{\text{val}}) := \sum_{m=1}^{M} \mathcal{I}(z_{\text{tr}}, z_m). \tag{58}$$

Incorporating with Equation (57), we get

$$\mathcal{I}(z_{\text{tr}}, S_{\text{val}}) := \sum_{m=1}^{M} \tilde{g}_{z_m} \tilde{H}_{\text{val}}^{-1} \tilde{g}_{z_{\text{tr}}} + \frac{1}{2}\epsilon \tilde{g}_{z_{\text{tr}}}^{\top} \tilde{H}_{\text{val}}^{-1} \nabla_{\tilde{\theta}}^2 \ell(z_{\text{val}}, \tilde{\theta}) \tilde{H}_{\text{val}}^{-1} \tilde{g}_{z_{\text{tr}}}$$

$$= \tilde{g}_{\text{val}} \tilde{H}_{\text{val}}^{-1} \tilde{g}_{z_{\text{tr}}} + \frac{1}{2}\epsilon \tilde{g}_{z_{\text{tr}}}^{\top} \tilde{H}_{\text{val}}^{-1} \tilde{g}_{z_{\text{tr}}}. \tag{59}$$

Since $\tilde{g}_{\text{val}} \to 0$, we define the Influence Function on the validation set $S_{\text{val}}$ as follows:

$$\mathcal{I}(z_{\text{tr}}, S_{\text{val}}) := \tilde{g}_{z_{\text{tr}}}^{\top} \tilde{H}_{\text{val}}^{-1} \tilde{g}_{z_{\text{tr}}}, \tag{60}$$

where $\frac{1}{2}\epsilon > 0$ is dropped.

## F. Related Works

### F.1. Influence Function

The use of influence-based diagnostics dates back to seminal works such as Cook & Weisberg (1982). More recently, Koh & Liang (2017) introduced Influence Functions to large-scale Deep Neural Networks (DNNs), sparking significant research interest. However, subsequent studies have identified challenges in applying IF to DNNs. Basu et al. (2021) highlighted the fragility of IF in deep learning scenarios, while Bae et al. (2022) demonstrated that practical influence function estimates often fail to align with leave-one-out retraining in nonlinear networks. Instead, they approximate a different quantity known as the proximal Bregman response function (PBRF).

Still, IF has inspired a wave of research aimed at improving its computational efficiency and scalability for large models. TracIn (Pruthi et al., 2020) improves computational efficiency by tracking loss changes using gradient information from multiple training checkpoints. FastIF (Guo et al., 2021) accelerates influence estimation by selecting the $k$-nearest training samples around a given validation sample. Schioppa et al. (2022) speed up the inverse Hessian calculation based on Arnoldi iteration (ARNOLDI, 1951). Grosse et al. (2023) proposed Eigenvalue-Corrected Kronecker-Factored Approximate Curvature (EK-FAC) to scale Influence Function computations to large language models (LLMs). GEX (Kim et al., 2023) alleviates the bilinear constraint of standard Influence Functions and leverages Geometric Ensemble (GE) for more precise influence estimation. DataInf (Kwon et al., 2024) further enhanced efficiency by deriving a closed-form influence estimation, making it particularly effective for parameter-efficient fine-tuning techniques such as LoRA (Hu et al., 2022).

Despite these advancements, previous studies have overlooked a fundamental issue: the relationship between influence estimation and the local sharpness of the validation risk. Regardless of how efficiently IF is computed, the absence of local sharpness correction results in inaccurate influence estimation, limiting its reliability in practice.

### F.2. Sharpness-Aware Minimization

The relationship between the flatness of local minima and generalization has been extensively explored in prior works (Dinh et al., 2017; Li et al., 2018). More recently, numerous studies have aimed to improve model generalization by explicitly seeking flatter minima (Chaudhari et al., 2017; Tsuzuku et al., 2020; Foret et al., 2021). Among these, Sharpness-Aware Minimization (SAM) (Foret et al., 2021) has attracted considerable attention. SAM introduces a general training framework that formulates optimization as a min-max problem, encouraging model parameters to lie in neighborhoods with uniformly low loss, thereby achieving state-of-the-art generalization across a wide range of tasks. Building on SAM, subsequent works have advanced the approach by refining the geometric definition of the neighborhood (Kwon et al., 2021; Kim et al., 2022), proposing improved surrogate loss functions (Zhuang et al., 2022), developing more adaptive adversarial strategies (Li & Giannakis, 2023), and improving training efficiency (Jiang et al., 2023).

# G. Miscellanies

## G.1. Remark on the Sign of Influence

In previous studies, the same training sample can yield positive or negative influence depending on the paper, due to differing definitions of loss change. This ambiguity arises particularly between $\ell(z, \theta_z) - \ell(z, \theta)$ and $\ell(z, \theta) - \ell(z, \theta_z)$. For example, Koh & Liang (2017) define the influence as $\ell(z, \theta_z) - \ell(z, \theta)$, whereas Pruthi et al. (2020) define it as $\ell(z, \theta) - \ell(z, \theta_z)$. In this paper, we adopt the previous approach, measuring loss change as $\ell(z, \theta_z) - \ell(z, \theta)$.

