# OpenReview forum: "Towards Robust Influence Functions with Flat Validation Minima"
_ICML.cc/2025/Conference — ICML 2025 poster_

### Official Review · Reviewer_fYnW · 2025-02-21

**Overall Recommendation:** 3

**Summary:**

The article "Towards Robust Influence Functions with Flat Validation Minima" addresses the challenge of influence estimation in deep neural networks, particularly in the presence of noisy training data. The authors identify a fundamental limitation of existing influence function (IF) methods: their susceptibility to unreliable estimates due to the sharpness of the validation risk. To overcome this, they propose a novel approach that leverages flat validation minima for more accurate and robust influence estimation. This is achieved through a second-order approximation to minimize the impact of vanishing gradients and a refined parameter change estimation method tailored for flat test minima.

The paper evaluates the proposed methods (VM and FVM) across various tasks, including mislabeled sample detection, training sample relabeling, influential sample identification in text and image generation. The experimental results demonstrate superior performance compared to existing approaches, highlighting the importance of seeking flat minima for enhancing influence estimation accuracy.

**Claims And Evidence:**

Claim: Influence functions suffer from unreliability when applied to noisy training data due to the sharpness of validation risk.

The authors provide both theoretical analysis (e.g., Theorem 3.2 and Corollary 3.3) and empirical observations (e.g., Figure 1 and Figure 3). These demonstrate how sharp validation minima introduce gaps between estimated and actual influence, undermining reliability.

The combination of theory and experimentation effectively supports this claim.

**Essential References Not Discussed:**

I argue that the experiments of EK-FAC-based Influence function such as reference [1]  should be compared with the proposed method.

[1] Grosse R, Bae J, Anil C, et al. Studying large language model generalization with influence functions[J]. arXiv preprint arXiv:2308.03296, 2023.

**Experimental Designs Or Analyses:**

The experimental designs and analyses in the paper are generally well-structured and provide strong support for the claims made.

**Methods And Evaluation Criteria:**

The proposed methods (VM and FVM) and the evaluation criteria used in the paper are generally well-aligned with the problem at hand—improving influence estimation in deep neural networks, particularly for noisy datasets

**Other Comments Or Suggestions:**

line 215 influence estimation error's math symbol is incorrect.

**Other Strengths And Weaknesses:**

Other weakness:
1. Assuming that the diagonal elements dominate the empirical Fisher Information matrix is an overly simplified solution that may introduce more theoretical bias.

2. I'm not entirely sure if the $ \tilde{\theta}_{ztr}$ needs to be recomputed for every training sample z. If so, the computational cost would be too high.

**Questions For Authors:**

1. How to calculate the $ \tilde{\theta}_{ztr}$ in eq. (17)?

2. How to calculate the $ \tilde{g}_{ztr}$ in eq. (21)?

**Relation To Broader Scientific Literature:**

The key contributions of the paper are closely related to several important areas in machine learning research, including influence functions, robustness in deep learning, and optimization landscapes.

1) The authors identify that existing influence function (IF) methods often fail to provide reliable estimates in deep neural networks, particularly when applied to noisy training data. They attribute this failure to deficiencies in loss change estimation due to the sharpness of validation risk.
2) The paper establishes a theoretical connection between flat validation minima and accurate influence estimation, emphasizing the importance of optimizing for flat minima.
3) The proposed methods (VM and FVM) significantly outperform existing approaches in detecting mislabeled samples and relabeling tasks.

The key contribution is based on the work of " Sharpness-Aware Optimization".

**Theoretical Claims:**

Yes.
The theoretical proof is largely correct, but I believe that Theorem 3.2 neglects the sample size $M$ ( eq. 35) when using Hoeffding's inequality.

---

> ### Author Rebuttal · Authors · 2025-04-01
>
> We appreciate the useful comments of the reviewer. We will update our current draft to avoid any confusion.
>
> > [Q1 / Other Weakness 1] How to calculate the $\tilde{\theta} _ {z_{tr}}$
>
> We would like to clarify that it is unnecessary to compute $\tilde{\theta} _ {z_{tr}}$for each training sample. Specifically, we only need to fine-tune the pre-trained model parameters $\theta$ to obtain $\tilde{\theta}$. Once this is done, we can approximate the parameter change $\tilde{\theta} _ {z_{tr}} - \tilde{\theta}$ as outlined in Eq 18.
>
> > [Q2] How to calculate the $\tilde{g} _ {z_{tr}}$
>
> After reviewing our initial submission, we found that the definition on line 244 indeed contains an typo, which we suspect may have contributed to this misunderstanding. Specifically, $\tilde{g} _ {z_{tr}}$ is defined as $\tilde{g} _ {z_{tr}} = \nabla _ {\tilde{\theta}} \ell(z_{tr}, \tilde{\theta})$, as demonstrated in line 879 in the appendix. We will correct this in the revised version.
>
> In practice, we implement this using Automatic Differentiation in PyTorch.
>
> > [Theoretical Claims] Sample size $M$ in eq. 35
>
> We can confirm there is no $M$ in eq. 35. We directly apply the variance proxy for the whole random variable $\mathcal{I}(z, S_\text{val})$ in eq. 35.
>
> We thank you for the observation, and will improve the readbility of the proof therein.
>
> > [Supp. Material] Code for Generation Tasks
>
> We promise we will make the code for reproducing all reported experiments publicly available upon acceptance.
>
> > [References] Comparison with EK-FAC-based Influence function
>
> Thank you for pointing out this important baseline that we had overlooked. We have now conducted a comparison with EK-FAC [r6] on mislabel detection tasks. The results (ROC AUC/AP) are reported below, where \* denotes the results computed using the best training checkpoints. As observed, our proposed method consistently outperforms the EK-FAC-based IF.
>
> |Method|C-10N Aggre|C-10N Random|C-10N Worst|C-100N Noisy|
> |-|-|-|-|-|
> |EK-FAC|57.03/39.71|69.85/57.70|72.18/72.81|60.30/59.37|
> |EK-FAC\*|70.02/60.51|81.55/76.58|80.24/81.72|62.72/64.86|
> |VM|95.18/76.31|95.92/87.35|95.88/94.27|89.77/83.81|
> |FVM|96.14/79.53|96.63/88.82|96.46/94.97|90.80/85.41|
>
> > [Other Weakness 1] Diagonal approximation for inverse Hessian
>
> Thank you for raising the concern regarding the potential oversimplification of our inverse Hessian approximation. Please refer to our detailed response to Reviewer tK1F for a full discussion on this point.
>
> > [C1] Typo
>
> Thank you for pointing out the incorrect mathematical symbol on line 215. We will correct this in the revised version.
>
> ---
>
> [r6] Studying large language model generalization with influence functions, arxiv, 2023.

---

### Official Review · Reviewer_Uyxg · 2025-03-13

**Overall Recommendation:** 4

**Summary:**

In this paper, the authors propose a method for estimating influence functions (IFs) for deep neural networks, addressing limitations of existing approaches that struggle with noisy training data and rely on first-order IF approximations without considering the sharpness of validation risk. They demonstrate that the error in the influence function estimator is directly related to the risk on validation samples. As a result, their IF estimator relies on parameters optimized for the validation risk. Additionally, the authors introduce a second-order approximation method tailored to their flat validation minima framework. Extensive experiments are conducted across various benchmarks and architectures.

**Claims And Evidence:**

Yes, the claims are supported by relevant evidence.

**Essential References Not Discussed:**

N/A

**Experimental Designs Or Analyses:**

Yes, the experimental setup and design are sound in most areas. However, I recommend that the authors update the manuscript to include clearer descriptions of the tasks and provide more detailed explanations of how each section of the experiments is carried out.

**Methods And Evaluation Criteria:**

Yes, the authors have considered relevant baselines and benchmarks datasets.

**Other Comments Or Suggestions:**

1. It would be beneficial to provide the experiment setup for Figure 2 either upfront or more explicitly in the caption.
2. The explanation for why removing $z_{tr}$ corresponds to $\epsilon = -\frac{1}{N}$ could be included in the preliminaries section.
3. **Minor**: In Figure 2, the blue legend line should be changed to "Validation ROC-AUC" to match the caption.
4. **Minor**: The title of subsection 3.2 needs to be updated.
5. $R_{val}$ in Theorem 3.2 and the corollary needs further explanation in the main paper.
6. In Corollary 3.3, Line 215, $\hat{\mathcal{R}}_S(\mathcal{I})$  should be replaced by  $\hat{\epsilon}_{S}(\mathcal{I})$  to maintain consistency (though I may be incorrect)
7. The experiment setup for Figure 3 should also be provided upfront for better readability.

**Other Strengths And Weaknesses:**

The proposed method is novel in its approach to addressing influence functions within the context of the flat validation minima framework that the authors consider. The experiment list is comprehensive, and the paper is well-written with minimal typos.

**Questions For Authors:**

1. The proposed approach for IF estimation depends on optimization with the validation set. Can the authors comment on how the performance might be affected by changes in the validation set size? Moreover, is there an assumption that this optimization will lead to an optimum that does not significantly deviate from the optimum obtained during the training phase, but instead moves toward regions where the validation loss is flat?

2. If the training data samples do not have noisy labels, will the method still provide the same performance benefits as the standard IF?

3. Since the method still requires computing second-order gradients to estimate the IF, can the authors comment on the computational complexities?

4. Can the authors explain Equation 23 in the paper and, in turn, provide a better explanation of the corresponding experiment? I reviewed Kwon et al., but I believe there is a major difference in your approach. Additionally, in Appendix B.2, only 10 validation examples are used for influence estimation. Could this pose a challenge for accurately estimating the IF?

**Relation To Broader Scientific Literature:**

In my humble opinion, the contributions are significant within the broader context of the scientific literature. However, there are some questions I have raised that require further clarification.

**Theoretical Claims:**

I did not go through the theoretical claims in great detail as I am not an expert in that. However, the overall arguments made logical sense and provided intuition.

---

> ### Author Rebuttal · Authors · 2025-04-01
>
> We sincerely appreciate the reviewer's valuable suggestion. After pondering over your questions, we compose the following responses and will include all the following elaborations in the revised manuscript.
>
> > [Q1.1] Validation set size
>
> We appreciate the reviewer for highlighting this important factor, and we will include a detailed discussion of this in the revision.
>
> Our approach consists of two steps: (1) fine-tuning and (2) influence function (IF) computation. In the second step, both our method and other IF-based approaches are similarly affected by the size of the validation set used for computing influence scores. Therefore, compared to other methods, the validation set size primarily impacts the fine-tuning step in our approach.
>
> To further investigate the impact of the validation set size on our approach, we conducted experiments comparing different IF approaches for mislabel detection on the CIFAR-10N Worst dataset with varying validation set sizes. The results (ROC AUC/AP) are reported below, with the performance drop relative to the size of 10.000 shown in parentheses.
>
> |Size|10000|5000|2000|1000|
> |-|-|-|-|-|
> |LiSSA|65.75/68.35|65.84/68.35 (+0.13%/-0.00%)|65.60/67.75 (-0.22%/-0.87%)|64.79/66.54 (-1.46%/-2.64%)|
> |VM|95.88/94.27|95.31/93.44 (-0.59%/-0.88%)|94.37/92.11 (-1.57%/-2.29%)|93.07/90.26 (-2.93%/-4.25%)|
> |FVM|96.46/94.97|95.68/93.71 (-0.80%/-1.32%)|94.86/92.60 (-1.65%/-2.49%)|93.72/90.94 (-2.84%/-4.24%)|
>
> As observed, the performance of all approaches tends to degrade. Nevertheless, our method maintains strong performance despite the reduced data size and still outperforms other approaches by a significant margin.
>
> > [Q1.2] Optimum assumption
>
> In Theorem 3.1, we assume that $\mathbb{E} _ {z \sim D_+}[\mathcal{I}(z, S_{val}) > 0]$ and $\mathbb{E} _ {z \sim D_-}[\mathcal{I}(z, S_{val}) < 0]$. As noted in the remark on line 201, this requires the overall influence estimation performance on $D$ (the training data distribution in practice) to be better than random guessing. This assumption implicitly relies on the training and validation data being drawn from related distributions and on the model retaining useful information about the training data. Consequently, we assume that the flat validation minimum does not deviate significantly from the given training minimum.
>
> > [Q2] Benefits under cases without noisy labels
>
> We would like to clarify that we have indeed conducted experiments in settings without noisy labels. Specifically, the generation tasks presented in Sections 4.3 and 4.4 are based on clean datasets, and our proposed methods, VM and FVM, consistently outperform existing baselines in identifying the most influential samples under these conditions.
>
> > [Q3] 2nd-order gradients and computational complexities
>
> In practice, to reduce both time and space complexities, we **approximate the Hessian** using the diagonal elements of the empirical Fisher Information Matrix, as described in Appendix A.
>
> Following the notation used in [r5], where $n$ and $m$ denotes the number of training samples and validation samples, $D$ the number of parameters per layer, and $L$ the number of layers, the computational complexity of LiSSA for estimating the Hessian inverse is $O(nD^2L)$, DataInf is $O(nDL)$ and our approximation is $O(mDL)$ (since it is performed w.r.t. the validation loss). Notably, in most practical scenarios, $m \ll n$.
>
> > [Q4.1] Experimental setting in generation tasks
>
> Regarding Equation (23), we acknowledge that its current form may have led to some misunderstanding due to limitations in presentation. We would like to clarify that the experimental setup for the generation tasks totally follows that of [r5]. Later, we will refine the explanation and notation of this part in the revision.
>
> > [Q4.2] Only 10 validation samples are used for influence estimation in LLMs scenarios
>
> We acknowledge the reviewer’s concern. Due to time constraints, we were unable to conduct more extensive experiments in the large language model (LLM) setting. However, we provide a simplified validation dataset size analysis on smaller-scale datasets—please refer to our response in [Q1.1] for details.
>
> We will include a more thorough discussion on this limitation and its potential implications in the revised version of the manuscript.
>
> > [C1-7] Writing
>
> We thank the reviewer for the detailed suggestions. We will revise the manuscript to improve clarity, including clearer descriptions of the experiment setups, explanation of why $\epsilon = -\frac{1}{N}$, and clarification of $R_{\text{val}}$, which refers to the risk on the validation set. The typo in Line 215 will be corrected. Additionally, we will also update the legend label in Figure 2 to "Validation ROC-AUC" and revise the caption of Figure 3 to include the experiment setups.
>
> ---
>
> [r5] DataInf: Efficiently Estimating Data Influence in LoRA-tuned LLMs and Diffusion Models, ICLR 2024.

---

### Official Review · Reviewer_r2Jq · 2025-03-15

**Overall Recommendation:** 4

**Summary:**

The influence function measures the influence of training samples on the validation loss. While this is typically done using minimas of the training loss, the authors argue for using flat validation minima. They then show experimntally and arguen theoretically that the standard estimators for influence do not work well in this setting and propose a new way of calulcating the influence that is designed to deal with these problems.

**Claims And Evidence:**

The basic claims are well supported, both theoretically and experimentally. I also think that the flow of the manuscript (observation -> problem -> solution) is coherent and convincing. The experimental evidence for the superiority of their approach is sufficient and extensive.

**Essential References Not Discussed:**

Not that I know of.

**Experimental Designs Or Analyses:**

No.

**Methods And Evaluation Criteria:**

Yes.

**Other Comments Or Suggestions:**

None.

**Other Strengths And Weaknesses:**

There has been a lot of discussion around the concept of flat minima and their dependence on the parameterization of the loss function (see below). This seems to be important also for the current paper but the authors do not discuss it. I think it would be good to have a sentence or two on the impact of these issues on the proposed method.

[1] Dinh, Laurent, et al. "Sharp minima can generalize for deep nets." International Conference on Machine Learning. PMLR, 2017.
[2] Pittorino, Fabrizio, et al. "Deep networks on toroids: removing symmetries reveals the structure of flat regions in the landscape geometry." International Conference on Machine Learning. PMLR, 2022.

**Questions For Authors:**

See above.

**Relation To Broader Scientific Literature:**

The paper does not have a "Related literature" section which makes it hard for a non-expert reader to assess novelty. I would strongly suggest adding one, since many of the core ideas (flat minima etc.) have vast literature attached to them.

**Theoretical Claims:**

I did not check the theoretical claims in detail (especially in the appendix) but from my reading they seem to be consistent.

---

> ### Author Rebuttal · Authors · 2025-04-01
>
> We sincerely appreciate your recognition of the theoretical and experimental support for our claims.
>
> > [Broader Sci. Literature.1] Absence of Related Literature section
>
> We kindly note that, due to space constraints, the discussion of related work on influence functions was not included in the main paper but has been deferred to **Appendix E**.
>
> > [Broader Sci. Literature.2] Further discussion on related works of flat minima
>
> We thank the reviewer for highlighting the importance of further discussing flat minima in the related works section.
>
> In the initial version, we acknowledge that we did not elaborate on prior work related to flat minima. This decision was based on the observation that most existing literature focuses on flat minima in the context of model generalization. In contrast, our work considers flat validation minima as a key factor for accurate IF estimation. Given this shift in focus, we initially chose not to include a detailed discussion of the broader flat minima literature.
>
> Nonetheless, we agree that adding this context will help clarify our perspective and position our work more clearly. We will therefore incorporate a more comprehensive discussion of relevant literature in the revised version.
>
> > [Other Weakness] Impact of "parameterization of the loss function"
>
> First, we would like to highlight that the main theoretical contribution of our manuscript is to establish a connection between flat validation minima and the accuracy of influence function estimation. The focus of our analysis is on **how flat minima matter for IF**, rather than on how to obtain them.
>
> To this end, we adopted a commonly used method—Sharpness-Aware Minimization (SAM)—to obtain flat validation minima, primarily as a means to support our theoretical insights rather than as a central contribution of the paper.
>
> We sincerely thank the reviewer for suggesting relevant literature [1][2] on achieving flat minima from the parameterization of the model. While these directions are indeed insightful, due to time constraints and the lack of public code for those papers, we opted to experiment with two more recent and open-sourced sharpness-aware optimizers as alternatives.
>
> Specifically, we compare three different sharpness-aware optimizer, including SAM (utilized in our initial submission), ASAM [r3] and F-SAM [r4], on CIFAR-10N Worst and CIFAR-100N Noisy dataset. The results (ROC AUC) are presented below.
>
> |Method|CIFAR-10N Worst|CIFAR-100N Noisy|
> |-|-|-|
> |SAM|96.46/94.97|90.80/85.41|
> |ASAM|96.71/95.11|90.83/85.25|
> |F-SAM|96.76/95.29|91.25/86.06|
>
> As observed, different sharpness-aware optimizers do affect the final results, with F-SAM achieving the best performance. This supports our hypothesis that better flat validation minima can lead to more accurate influence function estimation.
>
> ---
>
> [r3] ASAM: Adaptive Sharpness-Aware Minimization for Scale-Invariant Learning of Deep Neural Networks, ICML, 2021
>
> [r4] Friendly Sharpness-Aware Minimization, CVPR, 2024

---

### Official Review · Reviewer_tK1F · 2025-03-25

**Overall Recommendation:** 3

**Summary:**

This paper reexamines influence functions (IF) in deep learning and argues that their standard formulations fail when models are trained on noisy data—primarily because of sharp validation risk landscapes. The authors theoretically link the estimation error of influence functions to the sharpness of the validation loss and posit that obtaining flat validation minima (via techniques such as Sharpness-Aware Minimization) is key for accurate influence estimation. To that end, they propose a new influence estimator based on second‐order approximations for both the parameter change and the loss change, tailored specifically for flat validation minima. Extensive experiments on tasks such as mislabeled sample detection, training sample relabeling, and identifying influential samples in text and image generation are presented to demonstrate that their methods (denoted as VM and FVM) outperform existing approaches.

## update after rebuttal

The author's rebuttal addressed part of my concern, that the assumption of the Hessian being positive semi-definite is not part of their conclusion. I accept this reason.

**Claims And Evidence:**

The paper claims that the sharpness of the validation risk degrades the performance of standard influence functions and that flat minima lead to more reliable influence estimates. Although the authors provide theoretical bounds (Theorem 3.2 and Corollary 3.3) and empirical results, a critical issue arises with one of the core components: the inversion of the Hessian matrix. The diagonal approximation and its justification are not convincing in Appendix A. This weakens the overall support for the paper’s claims, as a robust inverse Hessian approximation is vital for the accuracy of the influence estimation.

**Essential References Not Discussed:**

N/A

**Experimental Designs Or Analyses:**

The experimental evaluation spans multiple tasks, including mislabeled sample detection and influential sample identification in both text and image generation. However, the reliance on benchmark datasets like CIFAR-10N/CIFAR-100N and controlled experimental settings means that the impact of the unclear Hessian inversion is not fully explored.

**Methods And Evaluation Criteria:**

The proposed method employs a flatness-aware objective (inspired by SAM) to find flat validation minima and then computes influence using a second-order approximation. However, a major methodological concern is the unclear treatment of the inverse Hessian. In Appendix A, the authors attempt to justify an approximation for the inverse Hessian—crucial for computing the parameter change—but the derivation and assumptions (such as the diagonal dominance of the empirical Fisher Information matrix) are not convincing enough. This lack of clarity raises doubts about the stability and reliability of the proposed estimator.

**Other Comments Or Suggestions:**

The authors should provide a clearer, more detailed justification for their inverse Hessian approximation. Additional empirical or theoretical evidence validating the assumptions made in Appendix A would be valuable.

**Other Strengths And Weaknesses:**

Strengths:

- The paper presents an innovative theoretical framework linking flat validation minima to influence estimation error.

- It introduces a novel estimator that, in controlled experiments, appears to outperform several recent baselines.


Weaknesses:

- The derivation and approximation of the inverse Hessian—a central element of the method—are not presented convincingly.

**Questions For Authors:**

See listed above.

**Relation To Broader Scientific Literature:**

The paper situates itself within a robust line of work on influence functions (e.g., Koh & Liang, 2017) and builds on recent advances by incorporating flat minima via SAM.

**Theoretical Claims:**

The paper's theoretical development is generally sound and well-structured. The authors derive bounds on the influence estimation error by explicitly connecting it to both the validation loss and its sharpness.

---

> ### Author Rebuttal · Authors · 2025-04-01
>
> We acknowledge the reviewer’s concern regarding the use of **diagonal approximation of the empirical Fisher matrix for estimating the inverse Hessian**. We are happy to provide a more thorough discussion of this aspect.
>
> ### 1. Diagonal approximation for inverse Hessian is simply a practical choice
>
> We would like to clarify that our core contribution lies in improving the influence function (IF) estimation by (1) **seeking flat validation minima**, and (2) **adapting IF computation to this scenarios**. Therefore, our core contributions are **orthogonal** to the specific choice of inverse Hessian approximation method. In fact, once flat validation minima are obtained, the specific method used to compute the inverse Hessian in Equations (20) or (21) becomes a matter of choice, depending on the trade-offs between efficiency and accuracy.
>
> ### 2. Further support for the use of the diagonal approximation
>
> We mainly refer to [r1] in JMLR, where the author discusses the diagonal of the empirical Fisher matrix and emphasizes that,
>
> - for accelerating 2nd-order optimization, the **diagonal approximation is a widely accepted method** to avoid the full computation of the Hessian [r2].
> - furthermore, optimization methods such as Adagrad and Adam, which are based on the Fisher matrix, also estimate second-order derivatives through an approximation to the diagonal of the empirical Fisher matrix [r1].
>
> These examples reflect a broader consensus in the community that approximating the inverse Hessian in DNNs through the diagonal of the empirical Fisher matrix strikes a practical balance between efficiency and effectiveness, especially in DNNs scenarios.
>
> ### 3. Additional experiments for different inverse Hessian approximation approaches
>
> We acknowledge that with a more accurate inverse Hessian approximation approach, the improved IF estimation performance can be expected.
>
> To quantitatively discover the impact of the approximation on the inverse Hessian, we conduct experiments on the mislabel detection task. Specifically, we replace the computation of the inverse Hessian $\\tilde{H}_{val}^{-1}$ in Equation 21 from diagonal Fisher to LiSSA and DataInf.
>
> > [Technical details in adaptation]
> >
> > Note that the accelerated approximations in LiSSA and DataInf both rely on the inverse Hessian-vector product (iHVP). However, the initial product $\\tilde{H}\_{val}^{-1} \\tilde{g}\_{z\_{tr}}$ in Equation (21) depends on the specific training sample $z_{tr}$, which requires recomputation for each training sample. As a result, we cannot directly apply these methods to our proposed influence function.
> >
> > To address this, we introduce a random vector $V \\in R^{|\\theta| \\times 1}$, where each element is sampled from a standard normal distribution, i.e., $V_i \\sim \\mathcal{N}(0, 1)$. With this, Equation (21) becomes $\\tilde{g}\_{z\_{tr}}^\\top \\tilde{H}\_{val}^{-1} V V^\\top \\tilde{g}\_{z\_{tr}}$, and we have $\\mathbb{E}[\\tilde{g}\_{z_{tr}}^\\top \\tilde{H}\_{val}^{-1} V V^\\top \\tilde{g}\_{z\_{tr}}] = \\tilde{g}\_{z\_{tr}}^\\top \\tilde{H}\_{val}^{-1} \\tilde{g}\_{z\_{tr}}$. By using the random vector $V$, we can directly apply the iHVP trick to $\\tilde{H}\_{val}^{-1} V$ and compute the inverse Hessian based on LiSSA or DataInf. In practice, we sample 5 different $V$ to ensure stability and reduce the variance of the approximation
>
> The results (ROC AUC/AP) are as follows:
>
> |Method|CIFAR-10N Aggre|CIFAR-10N Random|CIFAR-10N Worst|CIFAR-100N Noisy|
> |-|-|-|-|-|
> |VM(LiSSA)|95.22/67.36|95.67/78.90|95.00/89.26|88.88/80.89|
> |VM(DataInf)|95.37/72.09|95.98/83.26|95.57/91.09|89.58/82.81|
> |VM(ours, diagonal)|95.18/76.31|95.92/87.35|95.88/94.27|89.77/83.81|
> |FVM(LiSSA)|95.95/70.71|96.17/80.16|95.33/89.56|89.48/81.81|
> |FVM(DataInf)|96.18/76.15|96.58/84.85|96.02/91.54|90.24/83.81|
> |FVM(ours, diagonal)|96.63/88.82|96.63/88.82|96.46/94.97|90.80/85.41|
>
> Theoretically, from the perspective of computational complexity and estimation fidelity, LiSSA and DataInf are expected to provide more accurate approximations of the inverse Hessian than diagonal approximation. However, in our current experiments, we observe that the performance of the diagonal approximation is competitive with, and in some cases even superior to, LiSSA and DataInf.
>
> One possible reason is that LiSSA and DataInf involve more sensitive hyperparameters (e.g., number of iterations, damping), and tuning them appropriately for each setting is non-trivial. Despite this, the diagonal approximation—which is significantly more efficient—achieves consistently strong performance across all datasets. This suggests that our use of the diagonal approximation, while simplistic, does not lead to a substantial degradation in influence estimation accuracy in practice.
>
> ---
>
> [r1] New Insights and Perspectives on the Natural Gradient Method, JMLR 2020.
>
> [r2] Deep learning via Hessian-free optimization, ICML 2010.

---

### Decision · Program_Chairs · 2025-05-01

**Decision:**

Accept (poster)

**Comment:**

This paper introduces a connection between influence function, validation loss performance, and sharpness. The findings are well-supported and well-articulated through both theoretical and empirical arguments. After the rebuttal, reviewers have reached the consensus regarding their assessments — Therefore, I suggest its acceptance. One reviewer mentioned that there’s another line of related work on measuring flatness based on the loss function Hessian (e.g., [1](https://arxiv.org/abs/2306.08553), [2](https://arxiv.org/abs/2206.02659)). We encourage the authors to include these related works in the revision and also clarify any other issues surfaced from the reviewers’ feedback.